# A 21,000 year record of fluorescent organic matter markers in the WAIS Divide ice core

Juliana D'Andrilli[1, 2,], Christine M. Foreman[1,2], Michael Sigl[3,*], John C. Priscu[4], and Joseph R. McConnell[3]

[1]Dept. Chemical and Biological Engineering, Montana State University, Bozeman, Montana, USA
[2]Center for Biofilm Engineering, Montana State University, Bozeman, 59717, USA
[3]Division of Hydrologic Science, Desert Research Institute, Reno, 89512, USA
[4]Dept. of Land Resources & Environmental Science, Montana State University, Bozeman, 59717, USA

*Correspondence to*: Juliana D'Andrilli (Juliana@montana.edu)

**Abstract.** Englacial ice contains a significant reservoir of organic material (OM), preserving a chronological record of materials from Earth's past. Here, we investigate if OM composition surveys in ice core research can provide paleoecological information on the dynamic nature of our Earth through time. Temporal trends in OM composition from the early Holocene extending back to the Last Glacial Maximum (LGM) of the West Antarctic Ice Sheet Divide (WD) ice core were measured by fluorescence spectroscopy. Multivariate parallel factor (PARAFAC) analysis is widely used to isolate the chemical components that best describe the observed variation across three-dimensional fluorescence spectroscopy (excitation emission matrices; EEMs) assays. Fluorescent OM markers, identified by PARAFAC modelling of the EEMs from the LGM (27.0-18,0 kyr BP; before present 1950), through the last deglaciation (LD; 18.0-11.5 kyr BP), to the mid-Holocene (11.5-6.0 kyr BP) provided evidence of different types of fluorescent OM composition and origin in the WD ice core over 21.0 kyr. Low excitation/emission wavelength fluorescent PARAFAC component one (C1), associated with chemical species similar to simple lignin phenols was the greatest contributor throughout the ice core, suggesting a strong signature of terrestrial OM in all climate periods. The component two (C2) OM marker, encompassed distinct variability in the LD describing chemical species similar to tannin- and phenylalanine-like material. Component three (C3), associated with humic-like terrestrial material further resistant to biodegradation, was only characteristic of the Holocene, suggesting that more complex organic polymers such as lignins or tannins may be an ecological marker of warmer climates. We suggest that fluorescent OM markers observed during the LGM were the result of greater continental dust loading of lignin precursor (monolignol) material in a drier climate, with lower marine influences when sea ice extent was higher, and continents had more expansive tundra cover. As the climate warmed, the record of OM markers in the WD ice core changed, reflecting shifts in carbon productivity as a result of global ecosystem response.

## 1 Introduction

In addition to its water stable isotope content that provides a proxy record of past temperatures (Dansgaard et al., 1993), ice archives atmospheric information on trace gases like $CO_2$ and $CH_4$ encapsulated in air bubbles and chemical species trapped in the ice lattice. Numerous inorganic species trapped in ice has been used to reconstruct past chemical compositions of the atmosphere, its recent change in response to growing human activities as well its past natural variability (Legrand and Mayewski, 1997; Petit et al., 1999; Johnsen et al., 2001; Alley, 2002; Wolff et

al., 2006; Jansen et al., 2007; Luthi et al., 2008). In contrast, as reviewed by Legrand et al. (2013), information on
the load and composition of the organic matter (OM) archived in deep ice are still very limited (Legrand et al., 2013).

Ice provides a unique environment for preserving microbes and other biological material (i.e. plant fragments, seeds, pollen grains, fungal spores, and OM (Priscu et al., 2007; Miteva et al., 2009; Barletta et al., 2012; WAIS Divide Project Members, 2015). OM, comprised of biomolecules from living and decaying organisms, and also input from surrounding environments, plays a significant role in aquatic ecosystems affecting many biogeochemical processes that, in turn, influence its contribution to the global carbon cycle (Battin et al., 2008). Englacial OM may also contribute to global carbon dynamics upon its decomposition after being released from the ice lattice by melting and retreat (Priscu and Christner, 2004; Priscu et al., 2008). Therefore, as a reservoir of chronologically preserved OM and a potential source of carbon and biological material, determining ice core OM composition and reactive nature is essential to understanding past carbon signatures that could impact our future.

Ice core studies rely on the paradigm that atmospheric deposition is the sole mechanism for specific gases and materials to become trapped in the ice. Extending that idea to include OM, the trapped material becomes a catalog of preserved paleoecological markers of Earth's history, which can be used to better understand biogeochemical processing, carbon stocks, and cycling events. While still a novel addition to deep ice core research, chemically characterizing the OM markers (i.e. composition and chemical species) of englacial ice is of particular interest for several reasons: 1) OM markers can be linked to its source (e.g., aquatic, terrestrial) describing different influences of past and present ecosystems, 2) OM markers can serve as a proxy for englacial biological activity from *in situ* production, potentially explaining anomalous concentrations of other gases (e.g., methane, carbon dioxide) in ice core research, and 3) OM could be a pivotal contributor to the global carbon cycle if materials released to surrounding environments are metabolized to greenhouse gasses (e.g., carbon dioxide and methane) in a warming climate.

We hypothesize that Antarctic englacial ice contains a chronological record of OM markers that reflect changes following the Last Glacial Maximum (LGM). To test this hypothesis we used fluorescence spectroscopy to generate Excitation Emission Matrices (EEMs), a bulk analytical method commonly used to probe OM source and nature in aquatic ecosystems. Fluorescence spectroscopy is advantageous to employ for a large sample set with low sample volumes due to rapid data acquisition and the wealth of information generated describing OM fluorescing markers, chemical character, and source influences (Coble et al., 1990; Coble, 1996; Stedmon and Cory, 2014). For this study, all possible sources of OM markers detected within the West Antarctic Ice Sheet (WAIS) Divide ice cores were considered, however we can only speculate on the possibility of *in situ* OM production due to methodological limitations. A total of 1,191 meltwater samples of limited volume (~7.5 mL) were examined from 1400 m of deep ice, corresponding to 21.0 kyr, extending from the LGM, through the last deglaciation (LD), to the mid-Holocene. This is the first high temporal resolution analysis of englacial OM markers by fluorescence spectroscopy from the Antarctic ice sheet.

## 2 Methods

### 2.1 WAIS Divide sample site description

Ice cores were collected as a part of the multidisciplinary WAISCORES project at 79.467 °S and 112.085 °W, Antarctica (Figure 1) (WAIS Divide Project Members, 2013, 2015). Snow precipitation at this site is relatively high with an average annual accumulation rate 0.207$m_{weq}$ $a^{-1}$ (Banta et al., 2008) , compared to other Antarctic locations, resulting in ice cores containing a high resolution record of trapped gases, chemicals, and biotic and abiotic constituents over the last 65,000 years (dating scale WDC06A-7 (WAIS Divide Project Members, 2013)).

### 2.2 Ice core collection, preparation, and melting

Ice core drilling and recovery was completed in 2012 to a depth of 3405 m, using a hydrocarbon-based drilling fluid (Isopar-K; ExxonMobil Chemicals). Ice cores were transported to the National Ice Core Laboratory (NICL) in Denver, Colorado, for ice core processing. For this project, 1400 m of ice cores (depths: 1300 – 2700 m below the surface) dating from 6.0 to 27.0 kyr BP [before present 1950] by the WDC06A-7 timescale (WAIS Divide Project Members, 2013)), were cut into 3x3x100 cm long sections and transferred to the Desert Research Institute (DRI) in Reno, Nevada, for continuous melting (4.5 mL $min^{-1}$) in a closed continuous flow analysis (CFA) system (McConnell et al., 2002; McConnell et al., 2007; McConnell et al., 2014). The quality of the ice cores was excellent, well below the brittle ice zone without cracks (a section of ice containing a break in continuity, but still intact), and fractures (a section broken completely). Meltwater (7.5 mL for each sample) from the inner most section of the ice cores was directed to a discrete sample collector (Gilson 223), and dispensed into pre-fired (425 °C for 4 h) Teflon lined septa sealed amber glass vials, maintained at 4 °C to minimize volatilization and atmospheric exposure. Deionized water blanks were routinely analyzed through the CFA system to ensure a contaminant free environment. A Milli-Q water blank was also collected in the sealed amber glass vials for fluorescence analysis due to potential septa lid contamination. Any resulting fluorescence was subtracted from all EEMs prior to further analyses.

### 2.3 Ice core OM absorbance

Prior to fluorescence spectroscopy, absorbance spectra of WD core meltwater samples were collected from 190-1100 nm (UV-Vis spectral range) using a Genesys 10 Series (Thermo-Scientific) Spectrophotometer with a 1 cm path length cuvette and VISIONlite software. Obtaining UV-Vis absorbance spectra are necessary for the post-processing calculations of spectral corrections including primary and secondary inner filter effects (Acree et al., 1991; Tucker et al., 1992). Absorbance values at 254 nm ($A_{254}$) greater than 0.3 a.u. require dilution prior collecting the UV-Vis absorbance spectra and EEMs. WD ice core OM samples were optically transparent, with measured $A_{254}$ values well below 0.3 absorbance units (a.u.) after blank correction; consequently, no sample dilution prior to UV-Vis absorbance measurements and EEMs was required (Miller and McKnight, 2010; Miller et al., 2010). Spectra were blank corrected against purified water from a Milli-Q system each day. UV-Vis absorbance spectra were subsequently incorporated into the spectral corrections calculations for post-processing the EEMs data.

### 2.4 Fluorescence spectroscopy

EEMs were generated on a Horiba Jobin Yvon Fluoromax-4 Spectrofluorometer equipped with a Xenon lamp light source and a 1 cm path length quartz cuvette. Excitation (Ex) wavelengths were scanned from 240-450 nm in 10 nm intervals and emission (Em) was recorded between 300-560 nm in 2 nm increments. Data integration time was 0.25 s and data acquisition was carried out in signal/reference mode using a 5 nm bandpass on both Ex and Em monochromators, normalizing the fluorescence Em signal with the Ex light intensity. Data were normalized by the area under the Raman peak of a Milli-Q water sample each day at Ex = 350 nm and Em = 365-450 nm (e.g., maximum value 3.33877 x $10^5$ counts per second at 396 nm) based on previously reported ranges (Lawaetz and Stedmon, 2009). Post-processing of the fluorescence data was completed in MATLAB to generate 3D EEMs, which included sample corrections for our specific septa-lid/vial blank subtraction, and Raman and Rayleigh-Tyndall scattering following the smoothing procedures outlined in drEEM (Decomposition routines for Excitation Emission Matrices; v. 0.3.0) (Murphy et al., 2013).

EEMs were prepared for multivariate parallel factor (PARAFAC) analysis following a similar procedure previously outlined for sample classification, normalization, and subset selection (Cawley et al., 2012) to model the WD fluorescent OM character. This procedure was selected after failed attempts to validate modeled results of the entire EEMs data set, due to high percentages of outlier removal, noise/scattering interference, normalization effects, and low percentages of data fitted by each component producing high percentages of residual fluorescence. Briefly, EEMs were grouped by fluorescence into separate categories (relatively subjective categorization based on resolved fluorescence patterns; i.e. protein- and humic-like, scattering, etc.), and normalized within each category group to their maximum Em intensities to reduce the compensating effects that occur when normalizing samples over greatly varying fluorescence intensities. Using a randomization selection program, 20 samples were selected from each group for the representative subset of samples (n=140) for PARAFAC analysis. PARAFAC analysis continues to be widely used to decompose EEMs into individual OM fluorescent chemical components (Bro, 1997; Stedmon et al., 2003; Murphy et al., 2013). A three component PARAFAC model was generated for the subset of samples by drEEM and the N-way toolbox scripts in MATLAB under non-negativity constraints (Stedmon and Bro, 2008; Murphy et al., 2013). The three component model was validated by split half analysis with all of the components in the split model tests finding a match with a Tucker correlation coefficient > 0.95 (Murphy et al., 2013). The core consistency value was 97%, which was within the acceptable range suggested for robust PARAFAC modelling. Two and four component models were attempted, with a validation of the two component model, but a considerably lower core consistency value for the four component model. PARAFAC analysis beyond three components produced additional modelled results of noise, thus we were unable to validate a four component model. Therefore, the three component model was selected to best represent the entire data set and was used for further interpretation of our results.

To investigate how the PARAFAC model components would potentially shift based on climate periods, three separate PARAFAC models (LGM, LD, and Holocene) were also tested, which produced somewhat redundant results (specifically for components one and three; C1 and C3) to our three component PARAFAC model of the entire data set. With large groupings of outliers varying over different climate periods, these separate models were not appropriate tools to analyze statistical changes in all of fluorescing components over time. However, the

variation of the fluorescing regions comprising PARAFAC component two (C2) from the LGM to the Holocene

were captured by this method, thus those results are presented as qualitative comparative complements to the original model.

### 2.5 Elemental Analysis

Meltwater from the interior section of the ice core was also used for a broad range of elemental analyses (WAIS

Divide Project Members, 2013) including calcium (Ca) as an indicator of continental dust. From the CFA system, meltwater was directed through Teflon tubing to two Inductively Coupled Plasma Mass Spectrometers (ICPMS, Element 2 Thermo Scientific) located in an adjacent class 100 clean room for continuous trace element analysis (McConnell et al., 2007). Non-sea salt calcium (nssCa) concentrations were calculated following standard procedures from measured total concentrations of Ca using abundances in sea water and mean sediment (Bowen,

1979). Concentrations of sea salt sodium (ssNa) data from the LGM through deglaciation were previously reported and referenced in this work as a sea ice proxy throughout all climate periods (WAIS Divide Project Members, 2013).

### 3 Results

**3.1 Fluorescent OM markers in the WD core**

WD EEMs (1,191 samples covering 1400 m of ice core) dating from the LGM (27.0 – 18.0 kyr ago BP) to the mid Holocene (11.5 – 6.0 kyr BP) were analyzed to characterize the OM fluorescing components in ice. All samples contained low Ex/Em wavelength (240-270 nm / 300-350 nm) fluorescence characteristic of more easily altered material by microorganisms, representing fluorescent OM markers potentially of proteinaceous (Coble et al., 1990;

Coble et al., 1998), polycyclic aromatic hydrocarbon (PAH) (Ferretto et al., 2014), and simple phenol, tannin, or monolignol (Coble, 2014) origin. Fewer samples (2.5%) contained OM fluorescence at higher Ex/Em wavelengths (240-250 nm / 340-530 nm), characteristic of more humic-like markers of terrestrial plant/soil origin (Coble et al., 1990). Examples of low and high Ex/Em wavelength fluorescence can be seen in Supplement Figure 1a-c. Since OM is a complex mixture containing a broad range of molecules in potentially overlapping fluorescent regions, the

application of PARAFAC analysis was used to resolve the representative subset of samples into individual OM fluorescing components characterized by their Ex/Em maxima. The PARAFAC fluorescing components were analyzed to describe the chemical composition of the OM fluorescent markers, and the modelled results were then further interrogated to identify the contributions of each fluorescing component over the three different climate periods.

Three OM PARAFAC components were identified from the WD EEMs (fluorescing regions shown in Figure 2a, and Ex/Em wavelength loading scores shown in Supplemental Figure 2). PARAFAC component one (C1; Figure 2a, top) showed maximum fluorescence in a region analogous to the secondary fluorescence of fluorophore peak B (tyrosine-like, Ex: 240 nm and Em: 300 nm), typically associated with microbial processing in aquatic environments (Coble et al., 1990; Coble et al., 1998). Regions of fluorescence at such Ex/Em wavelengths are

commonly referred to as "protein-like" but overlap with fluorescence of other origins (Coble, 2014). However,

without the primary region of fluorescence associated with fluorophore peak B (tyrosine-like) at higher Ex/Em wavelengths, the OM fluorescent marker of C1 cannot be determined to be tyrosine-like material of microbial origin by this method. Rather, OM with similar Ex/Em wavelength fluorescence has been documented for simple phenols (e.g., tannins and monolignols) commonly detected in natural waters (Coble, 2014). Simple phenolic OM is

characteristically lower in molecular weight, aromaticity, and is considered to be more easily altered in the environment, as compared to more humic-like material (Coble, 2014). Thus, we report the chemical composition of WD OM in C1 to be most similar to monolignol chemical species, ubiquitously found in the environment as the precursors to lignin material detected in vascular plants. Once thought to be generated in the environment from tyrosine, the biosynthesis of monolignols actually originates from phenylalanine via multiple enzymatic reactions,

therefore sharing protein-like origin, but ultimately is chemically linked to vascular plants as a fluorescent OM marker (Wang et al., 2013).

PARAFAC component two (C2; Figure 2a, middle) contained maximum fluorescence at low Ex/Em wavelengths (260-270 nm / 310-320 nm) in regions analogous to the primary fluorescence of fluorophore peak B, and cresol (methylphenol), commonly known as the building blocks of tannins (Kraus et al., 2003) and the major

components of soil and aquatic humic OM (Tipping, 1986). Secondary fluorescence commonly detected for fluorophore peak B (tyrosine-like) was not observed for C2, and the combination of fluorescence from C1 and C2 do not yield the appropriate primary and secondary fluorescent trends commonly associated with tyrosine-like OM. Therefore, by this method, PARAFAC identified two distinct components, that may have protein-like similarities, but cannot be inherently linked to amino acid-like material and microbial origin. Thus, we determined that C2

fluorescence was characteristic of a combination of protein-like and tannin-like OM markers based on the regions of overlapping fluorescence by this method. Similarly to the chemical species reported for C1, the low Ex/Em wavelength fluorescence of C2 indicates OM markers with lower molecular weights, aromaticity, and chemical species that are more easily degraded in the environment by microorganisms (Coble, 2014). We acknowledge the relatively sharp shift in fluorescence intensity ~Em 310 nm across all Ex wavelengths (Supplement Figure 2b),

however could not remove this feature (Em spanning > 10nm) without compromising the integrity of the component in this model. Further evaluation of the variability potentially represented in this component is discussed in Section 3.2 as a function of climate.

Component three (C3, Figure 2a, bottom) displayed fluorescence commonly associated with more humic-like material. Two humic-like fluorescing regions were identified that comprised C3: fluorescence at 1) Ex/Em: 240-

260/380-460 nm, characteristic of fluorophore peak A, and 2) Ex/Em: 300-320/380-460 nm, characteristic of fluorophore peak C, commonly associated with terrestrial plant and/or soil origin (Coble, 1996; Marhaba et al., 2000). Fluorescent OM markers in this region are linked with chemical species having higher molecular weights aromatic nature, and are considered to be less easily altered by biodegradation in the environment as compared to more labile material (Coble et al., 1990; Cory and McKnight, 2005; Murphy et al., 2008; Balcarczyk et al., 2009;

Fellman et al., 2010). Humic-like material encompasses both OM produced by soils and from the oxidation of gaseous organic precursors emitted by the continental biosphere (signatures of vegetation) (Coble, 1996). Unfortunately, we cannot determine the absolute soil versus plant origin by fluorescence spectroscopy, thus can only

consider the source of C3 to be a terrestrial soil/plant signature. While commonly referred to as the "more recalcitrant" fraction of fluorescent OM, studies have shown that terrestrial humic-like material is susceptible to photodegradation, therefore should not be considered as an unalterable fraction of OM (Osburn et al., 2001; Stedmon et al., 2007).

WD ice core OM PARAFAC components were uploaded to the OpenFluor database to compare and contrast C1, C2, and C3 with other environmental OM marker studies, however, no component matches were determined (Murphy et al., 2014). The OpenFluor database is a repository of a selection of samples, and while still growing to encompass a thorough library of fluorescent OM markers from highly variable environments, it is reasonable to expect non-matching results based on database queries. Our results matched no previously identified PARAFAC components uploaded to the database, which we attribute to the unique scope of this work and the great volume of samples spanning 21.0 kyr from Antarctic ice.

**3.2 Fluorescent OM marker contributions as a function of climate**

The fluorescent intensity percentages of the components (OM marker contributions) varied not only by composition but also throughout the 21.0 kyr record (Figure 2b). Average percentages of fluorescence intensities for each component are represented as a function of climate (LGM, LD, and Holocene) as grey dashed lines in Figure 2b. C1 depicts the only OM marker with the greatest contributions for the LGM comparatively, describing a dominance of monolignol OM chemical species over time. In contrast, C2 and C3 OM marker contributions were lowest for the LGM, describing chemical species associated with these fluorescent regions being more prevalent in the environment or more effectively transported to WD in warmer climates. Similar average percentages were reported for C2 in the LD and the Holocene (23.52 and 22.09 %), however the standard deviations for both climate periods were highly variable (17.45 and 21.49), as a result of the varying fluorescent intensities, thus no discernable trend based on the average contributions was deduced. C3 contributions were the lowest throughout all climate periods and it is important to note that even though average contributions are presented for the LGM and LD, no observed resolved fluorophores were detected, i.e. OM markers characteristic of fluorophore peaks A and C were representative of the Holocene only.

Three separate PARAFAC models, categorized by climate period, were subsequently generated to identify imperceptible component variation potentially masked by the original model (Supplement Figures 3 and 4). For these models C1 was found to be identical for all climate periods using this technique, and since C3 was only characteristic of the Holocene, comparing these models in this manner elucidated the variation in C2 over time (Figure 3a-c; LGM-C2, LD-C2, and Holocene-C2). While all C2 fluorescence was situated in the same low Ex/Em wavelength region (250-300 nm / 300-350nm) for all climate periods, the breadth and maxima fluorescent regions shift, potentially describing different types of fluorescing material over time. LGM-C2 maximum fluorescence is detected at shorter Em wavelengths for the LGM (Figure 3a), compared to the LD (Figure 3b), corresponding to a shift to OM chemical species with higher molecular weights and aromaticity, comparatively. Although quite similar, LGM-C2 and LD-C2 resulted from different fluorescing composition, suggesting each climate period contains unique material preserved in ice cores. Following the LD, Holocene-C2 is noticeably unique compared to the LD

(Figure 3c), and shared overlapping regions of fluorescence with phenylalanine (Teale and Weber, 1957). A shift to shorter Em wavelengths (blue shift) is commonly associated with the opposite trend of a red shift, i.e. indicating chemical species at lower molecular weights and aromaticity. Thus our results indicate that C2 OM markers of the original PARAFAC model (Figure 2a) and the supplementary PARAFAC models (C2 OM markers across all climate periods combined in Figure 3a-c) describe chemical species of tannin-like and protein-like origin, with more

protein-like influence in the youngest ice.

### 3.3 Trace element concentrations

The extent of terrestrial dust contributions to the fluorescent OM markers in the WD ice core was explored by analyzing the concentrations of elemental nssCa, commonly used to reconstruct past atmospheric composition in

paleoclimate research. Figure 4 shows the concentrations of nssCa together with the WD ice core co-registered $\delta^{18}O$ temperature record for reference (Marcott et al., 2014). The highest concentrations of nssCa were observed during the LGM (Figure 4), indicating greater dust loads to Antarctica in the older ice. The transition between the LGM and the LD was characterized by a decrease in nssCa concentrations over 2.0 kyr, followed by concentrations that, on average, remain considerably lower than reported for the LGM, results that were consistent with other Arctic and

Antarctic paleodust records from ice cores (Albani et al., 2016). It is important to note no direct comparisons between dust concentrations and OM qualitative markers or concentrations can be made with these data, as that was beyond the scope of this work. Rather, this information was subsequently utilized as discussion points to infer more information regarding the OM marker origin detected in the WD ice core.

### 4 Discussion

### 4.1 Deep englacial OM nature and origin

The worlds glaciers and ice sheets are believed to hold nearly six petagrams of carbon (Hood et al., 2015), representing a significant component of the global carbon cycle. Ice environments function as sinks of allochthonous OM by atmospheric deposition and aeolian transport (Stubbins et al., 2012), yet our understanding of the OM source

and its reactivity in these reservoirs, especially in deep ice, is in its infancy. We applied fluorescence spectroscopy to determine the climate specific differences in OM markers and reactive nature throughout time. The composition and chemical origins associated with PARAFAC components C1, C2, and C3 provided a bulk level representation of the terrestrial OM markers preserved throughout 21.0 kyr and initiated the foundation for future research. The contributions of fluorescent OM markers represented by PARAFAC component C1 were dominant throughout the

LGM, LD, and the Holocene compared to C2 and C3, indicating a consistent record of OM with chemical species similar to monolignols over time. As precursors to lignin-like polymers, C1 OM markers in the WD ice core represent the continual presence of terrestrially produced material effectively transported to West Antarctica. Decreasing contributions of C1 OM markers with increasing contributions of C2 and C3 suggest that more complex terrestrially derived OM is a function of ecosystem changes in a warming climate. We continue to acknowledge the

fluorescence overlap at low Ex/Em wavelengths (C1 and C2) with regions of fluorescence that describe "protein-like" material characteristic of microbial processes (*in situ* OM production and transformation), however cannot

confirm OM microbial origin by fluorescence spectroscopy alone. Miteva et al. (2016) reported that the presence of microorganisms in deep ice cores also suggests the possibility of *in situ* OM processing, which could have important implications for gaseous climate records (Rhodes et al., 2013; Miteva et al., 2016; Rhodes et al., 2016). Including *in situ* biological OM transformations in ice core research was recently proposed as an alternate mechanism for $CH_4$ production in ice from firn layers of the WD ice core (Rhodes et al., 2016). At this juncture, our bulk level fluorescent results cannot argue that *in situ* OM production is a major contribution to the OM markers detected by EEMs in the WD ice core due to the lack of resolved tyrosine- and tryptophan-like fluorescing components. To test specifically for such signatures, further research linking microbial metabolism to available OM energy substrates in ice cores is required.

The humic-like fluorescent OM marker contributions were considerably higher in the youngest ice, suggesting lower abundances produced and/or ineffectively interacting with the atmosphere/transport mechanisms to Antarctica in colder climates. Resolved humic-like fluorophores (peaks A and C) were not detected in the older ice from the LD and LGM periods. The first appearance of resolved humic-like OM fluorophores was reported at 11.061 kyr BP, 500 years after the Holocene began. Considering the overall characterization of OM markers with terrestrial linkages in the WD ice core, we hypothesize that increasing temperatures beyond the LGM were associated with more expansive vegetation cover and increased production and degradation of complex OM in terrestrial environments.

**4.2 Continental dust in the WD ice core as an indicator of OM transport**

Concentrations of nssCa has been shown to be a valuable proxy for terrestrial crustal dust in paleoclimate ice core records (McConnell et al., 2007; Gornitz, 2009; Lambert et al., 2012). As such, it is plausible to envisage a link between the concentrations of nssCa and other transported materials influenced by aeolian deposition, (e.g., OM concentration and composition, microbial biomass, and pollen grains). The relationship between glacial cycles and atmospheric deposition of dust in Antarctica is largely discussed in ice core studies. We applied an assumption that common transport processes of dust and OM markers together could be hypothesized only if dust and OM originated from the same continental areas. Therefore, in this work, we merely speculate on the influence of dust concentrations and OM composition measurable by fluorescence spectroscopy. While organic C concentrations were not available for this study, we present the fluorescent OM markers measured concurrently with nssCa concentrations to estimate the potential strength and continental locale of various transport mechanisms as a discussion point potentially responsible for the specific types of fluorescing OM throughout history.

The LGM contained the highest concentrations of nssCa when annual snowfall deposition was low (WAIS Divide Project Members, 2013). We speculate that the larger nssCa concentrations measured during the LGM likely originated from increased local continental dust loading (of South America origin) as well as more efficient atmospheric transport. At the end of the LGM, nssCa concentrations declined around 18.0 – 16.0 kyr BP, near the beginning of the LD, reflecting a decrease in continental dust loading as the climate warmed. Throughout the LD and the Holocene, abrupt increases or spikes in nssCa concentrations were observed over shorter time scales, potentially representing other continental contributors (e.g., Australia) that may play a significant role. These

increases may be linked to other atmospheric events, emphasizing the plethora of mechanisms by which OM can be
transported to Antarctica.

**4.3 OM marker source fluctuations over time**

Fluctuations in ice core OM fluorescent markers may be driven by a multitude of variables, including: ecosystem productivity, changes in precipitation and accumulation due to temperature shifts, sea ice extent, wind patterns, fires,
and volcanic activity, most of which are in some way governed by the relative climate conditions. Sea ice extent was determined from the concentration of ssNa in the WD ice core (WAIS Divide Project Members, 2013). Higher concentrations of ssNa were associated with more extensive sea ice cover in colder climates, whereas decreased ssNa concentrations coincided with $\delta^{18}O$ enrichment, and implied less sea ice extent during warmer climates (WAIS Divide Project Members, 2013). Higher concentrations of crustal dust (of South American origin) were predicted to
also be a result of greater sea ice extent in the LGM, compared to lower transport when sea ice extent retreats (WAIS Divide Project Members, 2013). Changes in atmospheric circulation could also affect OM transport to WD and has been reported as a possible explanation for the decrease in ssNa concentrations at the end of the LGM (WAIS Divide Project Members, 2013). Concentrations of ssNa can thus serve as a proxy for multiple reconstructions, and may also be used to indicate more or less marine influences on the WD ice core OM markers.
We speculate that extensive sea ice cover would reduce the effect of marine influences on the WD ice core OM, compared to when the ocean is less covered.

As our WD OM marker record represents the first continuous ice core data set, we cannot directly compare our record with other ice core OM records to better understand C cycling of our past. Instead, we employ marine sediment records that described C production and cycling on the same relative dating scales (LGM to mid
Holocene). We attribute the qualitative fluorescent OM marker contributions in the WD ice core record to be a function of changing environmental influences over time. Three OM markers were identified to track the relative contributions of different types of fluorescing material in the LGM, LD, and the Holocene. The largest contributions of C1 were present in the LGM, indicating that C materials similar to monolignol species, potentially originating from South America, were prevalent signatures of a more ice covered Earth. Lower contributions of OM markers
similar to tannin- and humic-like material were observed in the LGM, describing lower abundances in the environment or less effective interaction with the atmosphere. At the onset of warming, the years in between the LGM and the Holocene (6.0 kyr) were a transitional period, encompassing a climate with rising temperatures and decreasing sea ice extent. The contributions of all three PARAFAC components shift from the LGM to the LD, towards decreased monolignol OM and greater influence of more tannin- and humic-like material. OM markers
(LGM-, LD-, and Holocene-C2) represented by the supplementary PARAFAC models (Figure 3a-c) captured C2 variation during the relatively rapid changes in climate. All three components identified as C2 had different Ex/Em maxima, indicating shifts to different OM markers characteristic of this transition time. The years between 13.0-11.5 kyr BP, at the end of the LD, defined as the Antarctic cold reversal, incorporate a depression of temperature, just prior to the early Holocene, where reports of Ca and dust concentrations increase (Delmonte et al., 2002). Also
measured in the WD ice core (increases in nssCa between 13.0 and 11.5 kry BP; Figure 4), we speculate that these

environmental fluctuations during the Antarctic cold reversal may also explain the fluorescent OM variation in the LD (Figure 3a-c). We submit that the fluctuations in C2 OM markers, observed in the LD, were a result of a multitude of environmental changes reflecting different C dynamics occurring after the LGM.

The Holocene marked the only period containing evidence of a more humic-like OM PARAFAC component (C3), albeit with relatively low percent contributions. Based on marine sediment records, the Holocene is characterized by higher levels of C productivity and vegetation cover (Ciais et al., 2012), with atmospheric temperatures and the potential for marine influences at their highest. With gross terrestrial C production estimated to be double that of the LGM (Ciais et al., 2012), it is reasonable to predict that the Holocene would contain the most heterogeneous mixture of OM, comparatively. With temperatures rising rapidly, Earth's atmosphere changed drastically in the Holocene, and atmospheric concentrations of $CO_2$ and $CH_4$ increased substantially (WAIS Divide Project Members, 2013, 2015) from natural processes. Reports of higher concentrations of $CO_2$ and $CH_4$ also suggests evidence of increased levels of C utilization and production in the Holocene, which agrees with the humic-like and potentially more degraded types of fluorescing OM markers observed in the youngest ice. Overall, the OM present in the WD ice core was dominated by materials fluorescing at low Ex/Em wavelengths, describing more easily altered chemical species upon exposure to surrounding environments if released from the ice lattice. Further research is necessary to measure the patterns of potential OM transformation pathways upon melting and retreat. Specifically, measuring the susceptibility of ice core OM to photodegradation and also microbial metabolism (respiration rates, $CO_2$ and/or $CH_4$ concentration accumulation), is a necessary next step to project how this material could impact atmospheric concentrations of greenhouse gases in a warming climate.

**5 Conclusions**

OM is a complex mixture of heterogeneous, polydisperse, and polycyclic molecules, the nature of which may result from multiple sources. Analyses of the WD OM markers by fluorescence spectroscopy allowed for the development of a series of interrelated climate and chemical records focused on understanding changes of C dynamics in global systems spanning 21.0 kyr of Earth's history. PARAFAC modelling of the WD ice core fluorescent OM markers identified components used to better understand ecological influences in a changing climate, providing more information on the types of C produced than is currently reported in marine sediment records. Simple phenolic fluorescent OM markers were dominant features in all WD samples over time, suggesting a strong terrestrial OM control of Earth's past ecosystems. More humic-like OM fluorescence, characteristic of more degraded material was only detected in the Holocene, at a time when temperatures were warmer, precipitation and accumulation was greater, C productivity was higher, and tundra ecosystems were less expansive, advancing the probability for effective interaction of organic materials with the atmosphere. OM fluorescent markers detected in the WD ice core within different climates may have fluctuated as a result of the diverse variables introduced as the atmosphere and C processing dynamics shifted over time. Taken together, the fluorescent OM markers in the WD ice core suggests that simple phenolic chemical species such as monolignols and cresols (precursors to lignins and tannins), prevailed in all climate periods, and were the greatest contributors to Earth's atmospheric compositions throughout history.

**Author contribution**

C. Foreman, J. D'Andrilli, J. Priscu, and J. McConnell designed the experiments and J. D'Andrilli and M. Sigl carried them out. Both J. D'Andrilli and M. Sigl were a part of the ice core melting team. J. McConnell and M. Sigl calculated the dating scale for all samples. J. D'Andrilli prepared the manuscript with contributions from all coauthors.

**Competing interests**

The authors declare that they have no conflicts of interest.

**Acknowledgements**

This work was supported by the National Science Foundation (NSF) Division of Antarctic Sciences through PLR-0839075, -0839093, and -1142166. We thank the West Antarctic Ice Sheet (WAIS) Divide Core community and field teams, participants of the WAIS Divide Meetings 2010-2013, the National Ice Core Laboratory (NICL), and members of the McConnell laboratory team who assisted in ice core melting in 2011-2012. Special thanks to G.R. Aiken, K. Hunt, J.R. Junker, D.M. McKnight, C.A. Stedmon, our anonymous reviewers, and A. Baker for their suggestions and contributions regarding organizing large dataset analyses and fluorescent interpretations. The authors appreciate the support of the WAIS Divide Science Coordination Office at the Desert Research Institute, Reno, NE, USA and the University of New Hampshire, USA, for the collection and distribution of the WAIS Divide ice cores and related tasks (NSF Grants 0230396, 0440817, 0944348, and 0944266). Kendrick Taylor led the field effort that collected the samples. The NSF Division of Polar Programs also funded the Ice Drilling Program Office (IDPO) and Ice Drilling Design and Operations (IDDO) group for coring activities; NICL for curation of the core; the Antarctic Support Contractor for logistics support in Antarctica; and the 109th New York Air Guard for airlift in Antarctica. Any opinions, findings, or conclusions expressed in this material are those of the authors and do not necessarily reflect the views of the NSF.

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

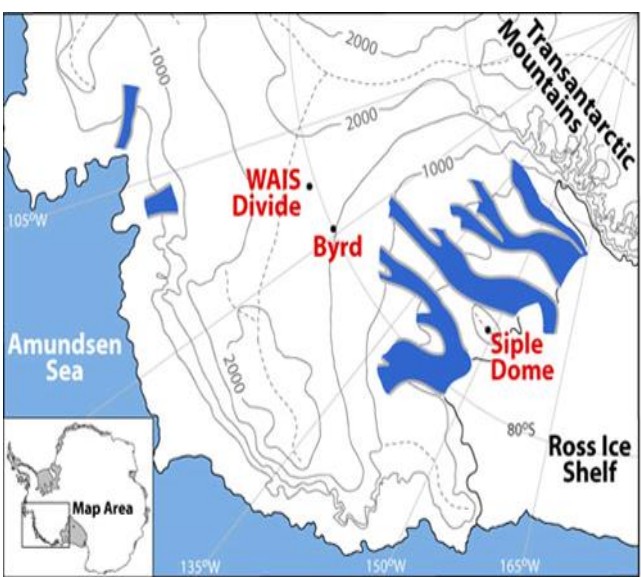

**Figure 1: Location of the West Antarctic Ice Sheet (WAIS) Divide in western Antarctica, with elevation contour lines: 112.085˚W Longitude, 79.467˚S Latitude, and 1,766m surface elevation (http://www.waisdivide.unh.edu/).**

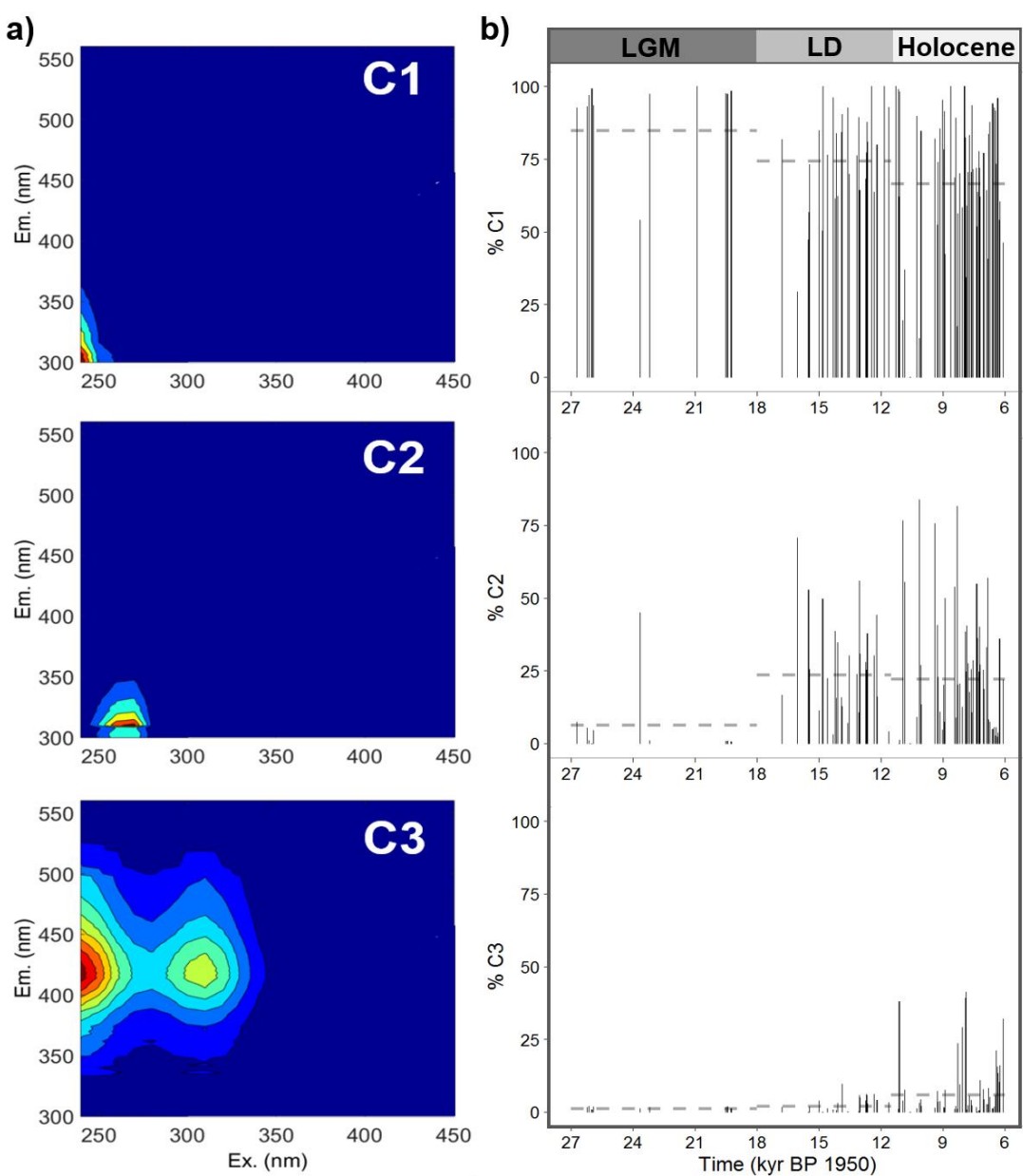

**Figure 2. PARAFAC analysis results for West Antarctic Ice Sheet Divide ice core organic matter showing a) components one, two, and three (C1, C2, and C3), and b) the fluorescence percentage of each component contributing to the overall fluorescence signature over the Last Glacial Maximum (LGM), last deglaciation (LD), and Holocene climate periods as a function of time (kyr before present 1950). Average fluorescence**

**percentages (gray dashed lines) are provided for each component, separately calculated for each climate period. Fluorescent data were reported in Raman Units. Note: C3 average fluorescence percentages are considerably lower in the LGM and LD, and did not correspond to resolved fluorophores.**

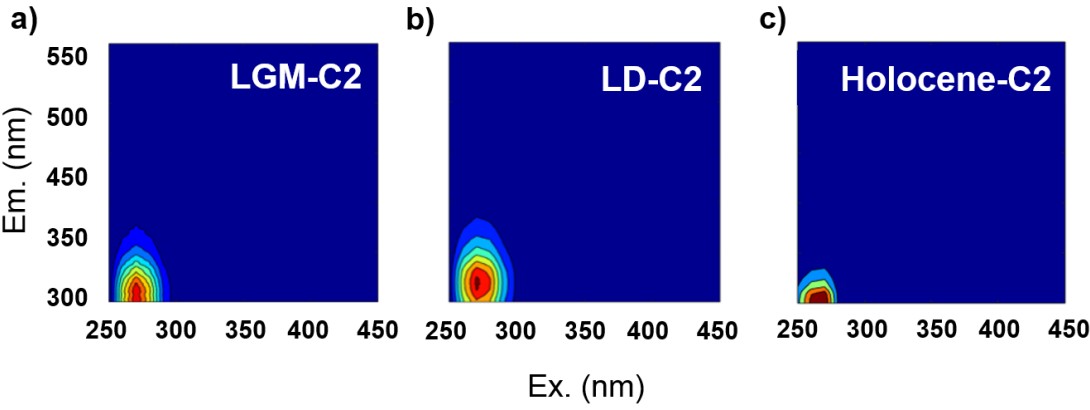

**Figure 3. West Antarctic Ice Sheet organic matter supplemental climate PARAFAC analysis combination results of component two (C2) variation with climate periods a) Last Glacial Maximum (LGM), b) last deglaciation (LD), and c) Holocene. Component one (C1) identified in each climate PARAFAC model showed no variability over time, and component three (C3) was only identified in the Holocene (Supplement Figure 3).**

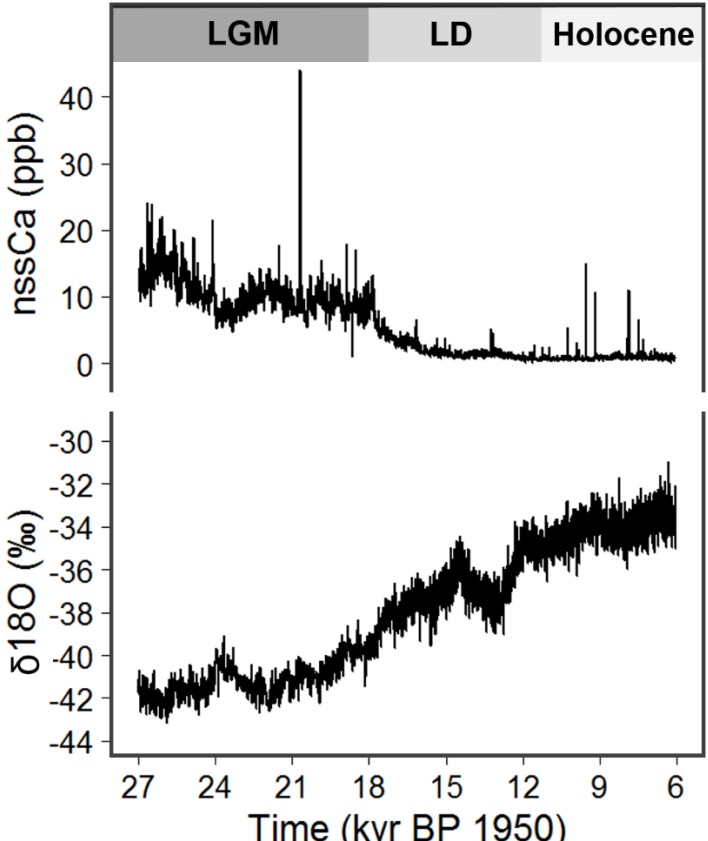

**Figure 4. Trace element concentration of (top) non-sea salt calcium (nssCa; ppb), and (bottom) the δ18O (per mil) temperature record (Marcott et al., 2014) from the West Antarctic Ice Sheet Divide ice core as a function of time (kyr before present 1950), dating from the Last Glacial Maximum (LGM), through the last deglaciation (LD), to the mid-Holocene.**

655