# Peer review of "A 21,000 year record of fluorescent organic matter markers in the WAIS Divide ice core"

_Climate of the Past, 2016_

## Referee Comment (RC1) · Anonymous Referee #1 · 3 Jan 2017

Comments to the paper J. D'Andrilli et al. A 21,000 year record of organic matter quality in the WAIS Divide ice core.

General Comments

The paper is concerning the stratigraphy of some organic matter markers (fluorescence signatures) along ice core sections from the WAIS Divide ice core (West Antarctica) covering the LGM, Last Transition and early-mid Holocene climatic periods. The topic is very interesting for ice core scientific community because measurements of organic matter (OM) or OM markers, as well as organic carbon, are very scarce in Antarctic ice cores and every new record, especially if obtained at high resolution, is useful in understanding the complex interactions between biological activity and climatic changes. However, the manuscript presents several weak points and, in my opinion, a deep re-

vision is necessary before it can be accepted for publication on Climate of the Past journal. The main criticisms are discussed in the "Specific and minor Comments" section end are concerning: 1. the results of the PARAFAC method; 2. the relationship between C1, C2 and C3 PARAFAC components with the different climatic periods; 3. the relationship between OM markers and dust deposition; 4. the relationship between the OM markers and volcanic activity. Besides, some methodological aspects have to be clarified. As a conclusion, in my opinion, the manuscript is not ready to be accepted for publication on CP in this form, but I'd like to encourage the Authors to re-submit a new improved version because their high-resolution measurements of OM markers are potentially very interesting in the paleoclimate studies.

Specific and minor comments.

Title and text. The term "organic matter quality" seems to be not adequate to describe the measurements here reported. Really, just fluorescence measurements were carried out and interpreted as signatures of some classes of organic components-like markers. I'd suggest the term "organic matter markers" or "organic fluorescent components".

Line 17 and several other points. Usually, time unit is expressed as "kyr" and not as "kyrs". Please, correct in the text and figures.

Lines 20-22. Here or in the "Results" section, Authors should clarify what They mean with the terms "labile microbial OM", "recalcitrant OM", "bioavailable carbon species" etc. A very short description of these terms could help the reader in better understanding the different biological significance and the different availability in carbon exchange between cryosphere and other ecosystems.

Line 32. Please, cite also Wolff et al., Southern Ocean sea-ice extent, productivity and iron flux over the past eight glacial cycles. Nature, 2006, Vol. 440, 491-496, doi:10.1038/nature04614.

Lines 51 and following. What "OM character" means? Chemical composition? Chemical-species or functional groups identification? Authors are requested to clarify their thought.

Line 55. Even methane formed in anaerobic conditions is a strong forcing factor in the warming climate.

Line 72. Since snow density is variable, it is better to express the mean accumulation rate as cm or mm "water equivalent".

Lines 74-76. What means this sentence? Several other ice cores (for instance, Taylor Dome and Talos Dome, in the same Antarctic Sector; Dome C and Dome Fuji, in the inner Antarctica; Dronning Maud Land, in the Atlantic Sector; etc.), even drilled before WD ice core, constitute "equivalent paleoclimate record" to Greenland ice cores. In particular, the EDC, EDML and DF climate records were compared with the climate oscillations recorded along the NGRIP ice core in: EPICA Community Members, One-to-one coupling of glacial climate variability in Greenland and Antarctica. Nature, 2006, Vol. 444, 195-198, doi:10.1038/nature05301.

Line 80. Please, change "drilling solvent" with "drilling fluid".

Line 88. Please change "combusted" with "pre-fired".

Section 2.3. The correction for the absorbance measurements seems to be not clear. Authors are asked to give more information on that. Besides, the absorbance threshold seems to be quite high. If a.u. means, as I think, absorbance unit, the value $A = 0.3$ corresponds to a percentage transmittance of 50% ($A = \text{Log } 1/T$) that seems to be too low for ice-core melted water at 254 nm. Maybe, some particles were suspended or some gas bubbles were present in the melted samples during the measurements. Authors are requested to clarify this point.

Line 97. Maybe the term "optically dilute" could be changed in "optically transparent" (but I do not think that this term is correct for T% = 50%).

Section 2.4. Even if a reference is cited, Authors are requested to give some basic information about the PARAFAC multivariate analysis.

Section 2.5. Authors should here anticipate why some elements were considered in this paper (e.g., nssCa as crustal marker, ssNa as sea spray indicator, nss-SO4 spikes to identify volcanic deposition signatures, etc.). Besides, more detail is requested in calculating the ss- and nss- fractions of Na, Ca and SO4. Since both Na and Ca can be related to two main sources (sea spray and dust), a four-equation system is necessary to calculate the ss- and nss- fractions (particular attention has to be put in evaluating ssNa during the LGM and nss-Ca during Holocene). Finally, which sea water ratio was used for the calculation of ss-SO4? Have the Authors used the SO4/Na seawater ratio of 0.25 or a lower value taking into account the possible contribution of frost flowers as sea salt source?

Lines 126 and 128. Authors are requested to shortly describe the characteristics of "bioavailable carbon species" and "more recalcitrant species".

Lines 129-131. This early Holocene peak of fluorescent mater is interesting, as well as the larger peak around 21-22 kyr BP. Authors do not discuss these two features in the temporal profile of the WD ice core. I'd like to know the Author interpretation on these large depositions of organic fluorescent compounds, even if as a tentative hypothesis. It should be very interesting to perform some qualitative analysis (e.g., by HPLC-MS measurements) on these samples in order to clarify the nature of the fluorescent compounds.

Lines 132 and following. I surely do not want to minimize the contribution of the PARAFAC analysis, but I have to note that the result of its application is quite basic. From Figure S1, the separation of the fluorescent bands at 420 nm Em and 300 nm Em is very clear even without any multi-parametric analysis. The only significant result is the identification of two fluorescent components C1 and C2 at short Em and Ex wavelength. However, the two components are just attributed to two large organic compound

classes (amino acid-like fluorescent compounds), without a more specific characterization. Besides, the C1 and C2 fluorescent components are not clearly differentiated in terms of biological origin: C1 is attributed to tyrosine-like fluorescent compounds associated to "microbial processing in aquatic environment", while C2 is described as a fluorescent signature overlapping "between tyrosine- and tryptophan-like" fluorescent compounds. At line 177-178, Authors just report that C2 containing tryptophan-like fluorescence could represent "intact dissolved proteins . . . . . . freshly derived from microorganisms". Authors are requested to better organize, in the present section, the discussion on the possible origin of these components and to enlighten the biological and environmental differences. In conclusion, the PARAFAC analysis seems to be not able to "resolve the representative subset of samples into individual OM fluorescing components", as the Authors assessed at lines 132-133. Even the comparison with the OpenFluor database components did not give significant matches (if I have well understood lines 153-155).

Lines 142-143. The terms "red/blue shifted to longer/shorter Em wavelengths" are repetitions. Please, change in "Em-wavelength red/blue shifted" or "shifted to longer/shorter Em wavelengths". Authors should clarify the statistical significance of these shifts (especially from LGM to LD) and anticipate the consequent biological meaning (especially from LGM-LD to Holocene). Besides, which is the meaning of the red or blue shifts? When blue (red) shift occurs, is the C2 component a marker of tyrosine-like (tryptophan-like) fluorescent compounds?

Section 3.2. The relationship between glacial cycles and atmospheric deposition of dust in Antarctica is a very relevant and largely discussed topic in ice core studies. Here, the Authors have to take for granted the inverse relationship between site temperature and dust deposition (by citing the most relevant references) and anticipate the discussion on the possible relationships among temperature, dust and biological activity (or OM transport efficiency), as revealed by the fluorescence temporal profile. At this purpose, Authors should choose the preferred dust indicator among the possible

dust markers measured along the WD ice core (nss-Ca, Mn and Sr), also basing on the correlations between the elements (lines 165-166).

Lines 174 and 176. Maybe, "throughout time" is better than "throughout history".

Line 198-200. Common transport processes of dust and OM could be hypothesized only if dust and OM originated from the same continental areas. In LGM, Southern South America was supposed to be the major dust source area for Antarctica. In LD and, especially, Holocene, even Australia could have played a significant role. Therefore, Authors implicitly suppose that MO was originated in these continental regions. For OM originated by marine sectors (C1, C2?, part of C3), the relationship with dust transport processes cannot be considered significant because they can follow very different pathways (e.g., implying different meridional or zonal atmospheric circulation modes).

Lines 200-201. Authors here refer on relationships between dissolved organic carbon and dust markers. I suppose DOC measurements were not performed as part of this paper (see following sentence in the text). Authors should give more information on that or cite some reference.

Line 204. I think Authors refer to Figure 4.

Lines 205-212. This part has to be completely revised. The complex relationship between dust deposition in Antarctic ice cores and climatic cycles cannot be discussed in this form in this paper and, how I have already pointed out, has been (and will be) the topic for several specific papers. Authors are requested to report the major literature references about LGM-LD-Holocene dust/climate pattern and focus the discussion on the relationship among climate, dust (possibly) and OM fluorescent markers. Besides, I have to note that the detail in the discussion on the behavior of OM data and dust profiles along the WD ice core is not so high to appreciate specific differences in nss-Ca, Mn and Sr profiles. Therefore, since the three dust-marker profiles were not singularly discussed and differentiated, I'd suggest to replot Figure 4 with just one dust marker

(maybe, nss-Ca).

Section 4.3. Even this section has to be largely revised. Authors assume a series of speculations to correlate changes of OM fluorescent markers to changes in climatic and environmental conditions, as evaluated by changes in sea-ice coverage (by ss-Na – Authors could add the ss-Na profile in figure 4), dust production and transport (by dust markers) and volcanic eruption frequency (by nss-SO4 spikes) in the LGM, LD and Holocene. However, no reliable comparison among the different time profiles is shown. In particular, while dust and sea ice markers show a progressive decreasing during the LD, the OM fluorescent profile shows an abrupt change (at about 18.5 kyr BP) from high LGM values and very low LD and Holocene levels. All the discussion is too elemental and also the changes in C1 and C2 relative contributions are not clearly interpreted. From the data here reported, I can just see that OM fluorescent markers are high in the LGM, when dust and sea spray are high. However, there is not experimental evidence on which climatic or environmental factors (more efficient meridional or zonal atmospheric transport, larger sea ice coverage, higher input from continental areas, larger emissions from marine biota, etc.) could have driven the OM deposition at the WD site. Finally, the relationship between volcanic activity (as recorded by the nss-SO4 spikes along the WD ice core) and OM fluorescent markers is, in my opinion, really unsustainable. Volcanic signatures in Antarctic ice core are mainly related to long-range atmospheric (especially stratospheric) transport of SO2 emitted during eruptions occurred at hemispheric scale and it is really difficult to correlate changes in WD OM to sporadic, short-time and widespread volcanic emissions without a strong experimental evidence.

Lines 225-226. What this sentence means? What is compared to the open ocean?

Lines 233-234. Authors are requested to better discuss the red shift of the C2 component, explaining which amino acid-like components increases its contribution to fluorescent OM and at which biological source can be attributed. What "external environments" means?

Lines 237-243. The pattern of the OM fluorescent markers during the ACR is not visible in Figure 3 (neither in Figure 2). This part is merely speculative and not supported by experimental evidences.

Lines 244-250. How can the Authors explain the very low levels of OM fluorescent markers during the Holocene, when climatic conditions should promote higher terrestrial and marine biological productivity? Which could be the significance of the large spike in OM fluorescent profile (Figure 2) at about 10 kyr BP?

Line 258. Please, change "Concentrations of nss-sulfur ..." with "Spikes in nss-SO4 concentrations ..."

Lines 261-262. Authors are requested to clarify how volcanic activity can stimulate OM production. How is calculated the percentage of the fluorescent OM attributed to the volcanic activity? The relationship between volcanic activity and OM deposition at WD site is, in my opinion, not plausible and not supported by experimental data (at least, by experimental data here reported). Have the Authors measured OM fluorescent peaks in ice core sections with volcanic depositions? In absence of experimental support, the discussion about the volcanic activity and OM fluorescent markers should be removed from the manuscript.

Conclusions section. This part should be changed accordingly to the changes suggested along the different manuscript sections.

---

## Referee Comment (RC2) · Anonymous Referee #2 · 4 Jan 2017

General comments:

The authors analyzed over one thousand samples for organic matter (OM) from the Last Glacial Maximum (LGM) until the uppermost section of the WAIS ice core. Through parallel factor multivariate analysis (PARAFAC) the authors determine three main components, where the third component (C3) comprises labile OM from terrestrial plants and soil and is only present during the Holocene. The other components (C1 and C2) are present from the LGM onwards. The authors compare their OM results to variations in terrestrial crustal dust input (nssCa, Sr, and Mn) to examine if major glacial-interglacial differences in transport and/or aridity affect the OM composition.

This research presents one of the first continuous studies of OM in an ice core over glacial-interglacial timescales. While the increased input of OM from terrestrial sources

during the Holocene is not surprising, this research quantifies this change. This paper can therefore serve as a foundation for many future studies investigating OM in polar ice cores. The authors understate any possible changes in atmospheric deposition and/or possible in-situ processes that may affect their results, but the current study may not have sufficient information for determining these changes.

Specific comments:

You state that "Ice core studies rely on the paradigm that atmospheric deposition is the sole mechanism for specific gases and materials to become trapped in the ice" (Lines 47- 48) yet it is unclear if you apply this paradigm to your work. If you do not allow even a remote chance for in-situ production of this organic matter, then please explicitly state so in your work. In lines 179-183 you mention the possibility of in situ OM processing but then do not discuss if such transformation could affect the samples in this work. You mention that tryptophan-like florescence in C2 may derive from microorganisms, and then mention that the presence of microorganisms may result in in situ OM processing, but step back from linking the two aspects. In the following paragraph you then mention that Holocene terrestrial plants and soils are the likely source of the C3 OM yet do not mention if in situ processes may affect this material or if you ascribe this material to be solely brought in via atmospheric transport. Please clarify your stance on the source and possible post-depositional processes affecting the samples as both aspects are essential to your interpretations of the data.

Please check that all figures are cited in the text. In lines 128-144 you mention Supplemental Figures 1a-b. You do not refer to Figure 2 in the text. As you refer to the Supplemental Figures but not Figure 2, then perhaps their roles should be reversed with the current Figure 2 included in the Supplementary Information and vice versa.

The left bars and corresponding explanation in the caption of Figure 3 are confusing. In the article text you explicitly state that C3 only occurs during the Holocene. As most readers will likely first look at the figures and captions before reading the article, it

bears mentioning in the caption that C3 is specific to the Holocene. Demonstrating the variation in C2 by various time periods (LGM, LD and Holocene) is useful but then makes the reader immediately wonder what is the variation in C1 between climate periods. If there is no substantial variation between time periods for C1, please mention this fact in the caption.

This sentence is confusing (Lines 227-229): "During the LGM, tundra ecosystems covered more expansive areas of the Earth (Ciais et al., 2012) and while C was cycling, productivity in the environment differed from warmer climates (Ciais et al., 2012 and references within)". Do you mean due to the colder temperatures and increased ice cover and tundra during the LGM, that net C productivity was less than in the other warmer times periods of this study?

The final conclusion overstates the results of the study. To state that labile, microbially derived OM "were the greatest contributors to Earth's atmospheric composition throughout history" is not correct. Labile OM may have been the greatest contributor of total OM in the atmosphere over the time periods covered in this paper, but this situation may not be the case before the LGM. In addition, in this sentence it is not clear what aspect of the "Earth's atmospheric composition" that you mean.

Technical corrections:

Line 16 = Define PARAFAC as this is the first time that you use this acronym.

Line 48: Place "idea" after "that" in "Extending that to include".

Line 222: Remove the comma after "LGM".

---

## Referee Comment (RC3) · Anonymous Referee #3 · 8 Jan 2017

The paper reports on the record of the fluorescence signature of the organic matter present in an ice core drilled in East Antarctica and covering the last great climate change having occurred between the last glacial maximum and the Holocene period. Very little is known on the level of organics in ice, especially in the case of Antarctica. Therefore these data are very interesting for researches dedicated to our understanding of the atmospheric budget of the organic matter and its change in response to past climatic conditions. However, major revisions of the manuscript are needed before I can recommend publication of this work in CP.

**The first weakness of the manuscript is the use of poorly defined wording rendering difficult (sometimes obscure) the reading of the manuscript.**
For instance, I guess that, when saying "OM quality", you mean "fluorescent signal of the OM" ? Also what is a recalcitrant OM ?

Some abbreviates appear in the text without definition. For instance, what is the PARAFAC model that is already mentioned in the abstract, also please indicate what is the basic of this kind of model ?

The abbreviates C1, C2 and C3: I guess that they refer to component 1 etc (and not to C1 carbone chain etc).

In section 2.3, please define A254 and re-define EEMs here.

Section 2.4: I don't understand the following sentence "A three component PARAFAC model was generated for the subset of samples by drEEM and the N-way toolbox scripts » : what is « drEEM » and N-way ?, please define.

Concerning units: Line 98 : what is au ?

I will avoid the use of RU for Raman unit (RU is sometimes used for relative unit). Also I am not sure that the readers of CP, specially those working on ice cores, are familiar with this Raman unit ? A few words on that would help (see also my comment on Figure 2).

**Introduction, first paragraph (lines 31-446):**
This paragraph can be improved significantly, for both the wording and the cited references. Two of your co-authors have a nice expertise on the chemistry of ice cores, they certainly can also help here.
From my side I would suggest to start with an overall sentence: "In addition to its water stable isotope content that provides a proxy record of past temperature (see Dansgaard et al. (1993), for instance), ice archives atmospheric information on trace gases like $CO_2$ and $CH_4$ encapsulated in air bubbles and chemical species trapped in the ice lattice. Numerous inorganic species trapped in ice has been used to reconstruct past chemical composition of the atmosphere, its recent change in response to growing human activities as well its past natural variability (see Legrand and Mayewski for a review)."

I here agree with another reviewer of the manuscript that the Nature paper from Wolff and co-workers (2006) is an excellent example that you have to mention of what was done on deep Antarctic ice cores in terms of changing sea-ice dust emission and marine biological productivity over the 8 climatic cycles.

Then focus on what was done on organics saying "In contrast, as reviewed by Legrand et al. (2013), information on the load and composition of the organic matter archived in ice are still very limited. »

I think you can find in this review paper relevant references that can be useful for your introduction. In particular, I suggest to report the work from Amanda Grannas made of the nature of OM in polar ice and those done on the HULIS like content of ice.

Legrand, M., and P. Mayewski, Glaciochemistry of polar ice cores: A review, *Reviews of Geophysics*, 35, 219-243, 1997.

Legrand, M., S. Preunkert, B. Jourdain, J. Guilhermet, X. Fain, I. Alekhina, and J.R. Petit, Water-soluble organic carbon in snow and ice deposited at Alpine, Greenland, and Antarctic sites: A critical review of available data and their atmospheric relevance, *Clim. Past*, 9, 2195-2211, doi:10.5194/cp-9-2195-2013, 2013.

Grannas, A., Shepson, P. B., and Filley, T. R.: Photochemistry and nature of organic matter in Arctic and Antarctic snow, Global Biogeochem. Cy., 18, GB1006, doi:10.1029/2003GB002133, 2004.

**Section 2.1**.: line 75: WD is not at all the first Antarctic ice record available for comparison with Greenland records. Please modify the text.

**Section 2.2**: line 86: what is the difference between cracks and fractures ?

**Section 2.5:** Please write a few sentences explaining why your choice was to show these inorganic species. Note that, as far as I know (and checking your fig 4), I see no reason to use three species (Mn, Sr, and Ca) for dust (except if you have in mind to discuss the ratio between the 3 in view to eventually highlight the source region, which seems not to be the case).

**Figure 2:** Are there any possibility of estimate from the Raman values how much is the concentration of OM ? Indeed, given the scarcity of data on organics, even an order of magnitude would be welcome here. From that and using a typical conversion factor OM/C you can estimate the TOC or DOC content of ice. Also I am surprised that the spikes shown in the fluorescence intensity during the LGM are not more commented in the text.

**Section 4:** I feel that this section has to be (at several) places deeply revisited:

Line 184-195: I assume that "Humic-like fluorescent OM" corresponds to Humic like substances observed in the atmosphere of many regions. If correct, did you consider these species as primary emitted (with soil particles for instance) or secondary produced from oxidation of gaseous organic precursors emitted by the continental biosphere (vegetation)? See also the discussion in their presence in ice in the following recent paper.

Guilhermet, J., S. Preunkert, D. Voisin, C. Baduel, and M. Legrand, Major 20[th] century changes of water-soluble HUmic LIke Substances (HULIS$_{WS}$) aerosol over Europe inferred from Alpine ice cores, *J. Geophys. Res. Atmos.*, 118, doi :10.1002/jgrd.50201, 2013.

Section 4.2: Your discussion on change of dust tracers is quite oversimplified and I would recommend you to revisit previous works done on this topic.

Lines 255-265: This discussion is from my point of view rather confusing.

It is incorrect to say that nssS concentrations are used to trace back volcanic eruptions. Only the narrow peaks of nssS are related to volcanic eruptions whereas the background nssS level in Antarctica originates in marine biogenic emissions (please revisit here the paper from Wolff et al., 2006 for instance). Also, I don't think that the wording of the following sentence makes sense "Therefore, volcanic eruptions increase the potential for particles and chemicals to be transported to polar regions and deposited onto ice- sheets." Please modify.

**Supplementary material:** Following your line 201 on a correlation between DOC and nssCa, I checked the S2 figure (extracted below) that strongly bothers me. Indeed, if the DOC unit you report is correct, DOC levels of this Antarctic ice are as high as 200 $\mu$M. If I am right that means 12*200 $\mu$g L$^{-1}$ i.e. 2400 ppbC. If correct, please comment with respect to the review of Legrand et al. (CP, 2013). It is very likely that you have a large DOC contamination in this shallow WD core. Also, sorry but I don't see a good correlation in this figure between dust and DOC!!! Please comment.

[Figure]

**End of the review**

---

## Short Comment (SC1) · 20 Jan 2017

It is really interesting to see a record of organic matter fluorescence from an ice core. Organic matter fluorescence is in my expertise area, so I have provided these comments on the manuscript, focusing only on the optical analyses. I hope they are of use. I also recommend Coble PG et al 2014 Aquatic Organic Matter Fluorescence (Cambridge University Press) (Disclaimer – I am a co-editor and co-author of two chapters, but these chapters are not those I recommend here).

Line 88. When I have investigated the use of 'septa sealed vials', I find a contaminant fluorescent signal coming from the septa, which in my tests has always been fluorescent. Can the authors confirm that their septa sealed amber glass vials produced zero fluorescence blanks?

[Figure]

Line 89-90. Following on from my previous comment, were the blanks run just on the melting system, or the melting system and amber glass vials? It is not clear at present.

Line 97. What was the actual absorbance values? These should be plotted as a time series, as A254 is used as a surrogate for DOC in terrestrial systems. It would be interesting for the reader to see this data and for the authors to compare values to other terrestrial systems (e.g. rivers, groundwaters).

Line 106-107. Were the data also processed to remove Rayleigh-Tyndall scatter? How were the Raman and Rayleigh-Tyndall scatter lines processed? Were they replaced by zeros, by NaN (not a number) or was data interpolated? All of these effects can have subtle influence on the resultant PARAFAC model, so it is good to report them.

Lines 108-110. The authors must specify what they did for sample classification, normalisation and subset selection. It will be different from Cawley et al (2012), which is just one fluorescence case study, and on pulp mills, so not really very relevant to this research.

Lines 108-110. Somewhere in this section the authors must quote the value of the standard(s) that they were using. This could be the Raman intensity of Milli-Q water at a specific wavelength, or the intensity of quinine sulphate standards run using the same instrument configuration, or an International Humic Substances Standard, or a tryptophan or tyrosine standard.

Lines 110-111. More detail is needed on the PARAFAC model, to allow the reader to assess its strength in modelling the data. It is crucial in this paper, as the PARAFAC model is the crux of the whole analysis and interpretation. 1. One would expect to see the core consistency value given. A 'passable' model could be considered have a value of >90%, and a good model a score of >99%. 2. It would be very informative to know why the authors chose a 3 component model over a 2 or 4 component model – did the 4 component model try to model noise, for example? Or did it model a plausible 4th component, but with a low core consistency. 3. The percentage of the data fitted

by each component is very valuable information, especially if compared with that from a two and four component model. 4. And finally, a split-half analysis is very useful, especially if the authors perform a split half analysis using randomly split datasets and a split half analysis with LGM data in one dataset and Holocene data in the other. If the split half analysis fails on the latter test, then it tells you that the LGM and Holocene need different PARAFAC models.

Line 126-127. Amino-acid like fluorescence is too general. Only tryptophan and tyrosine have aromatic groups which fluoresce, and even then, without independent amino acid analyses to confirm their presence, one can never be sure that these compounds are responsible for the fluorescence. If the fluorescence is from an amino acids source, then C1 and C2 look most like a 'tryrosine-like' compound. Tyrosine would excite at both ∼225 nm and ∼275 nm and emit at about 310 nm. But the molecular structure is such that you must observe both the 225 and 275 nm excitation of the 310 nm emission, not just one or the other, as you show in Figure 3. Supplemental Figure 1 confirms the absence of a ∼275 nm excitation peak. Therefore C1 and C2 are not 'tyrosine-like' or 'tryptophan-like'. Model compounds and contaminants that exhibit a single peak in this general region include simple phenols such as cresol (see Aiken, 2014 in Coble et al. (eds) Aquatic Organic Matter Fluorescence), PAHs such as fluorene (Ferretto et al 2014, DOI:10.1016/j.chemosphere.2013.12.087) and aviation fuel (see Baker et al. 2014, Encyc. Anal. Chem. DOI: 10.1002/9780470027318.a9412).

Line 128. The reference to 'recalcitrant species' is speculative. It would be better to specify the excitation and emission wavelengths of this peak or peaks. I am not aware of fluorescence in this region being recalcitrant – instead bio- and photo- degradation studies show that it is degradable (for example, Osburn et al and Stedmon and Cory, both in Coble et al 2014).

Line 129 and Figure 2. There is almost no meaning in 'total OM fluorescence intensities'. Each fluorophore has a different fluorescence efficiency. For example, in this study, you identify three fluorescent components, but each will have a different amount

of emitted fluorescence per g C present. So, summing the three is meaningless. It is particularly relevant as low molecular weight compounds such as tryptophan-like and tyrosine-like compounds (argued to be C1 and C2 here) have less chance of their emitted fluorescence being reabsorbed within the molecule, and they therefore have relatively high fluorescence efficiency. In contract, fulvic-like compounds (arguably C3 here) can reabsorb their emitted fluorescence, resulting in a much lower fluorescence efficiency. Figure 2 is therefore just meaningless and instead each PARAFAC component score (C1, C2, C3) needs to be presented.

Line 134 and Figure 3. The PARAFAC scores for C1, C2 and C3 need to be presented in Figure 3. At the moment, no raw data from the PARAFAC model is presented in the paper, yet this is the main focus. The reader has no way of seeing the data and judging its nature e.g. variability over time. Just drawing some PARAFAC model EEMs over an x-y plot would be unacceptable to the fluorescent organic matter research community.

Line 134-136. This observation needs quantification (see comment above).

Line 137-139. As in my earlier comment, you cannot have just one of the two excitation peaks that 'tyrosine-like' compounds excite at, and then call it 'tyrosine-like'.

Line 139-145. There is most fluorescence at 310 nm, so this is not 'tryptophan-like' at all, as this would also have a peak at 350 nm. More fundamentally, there is a line through the EEM at 310nm which cannot be real. Is this an artefact of the design process of Figure 2, or is it in the actual PARAFAC model? If the latter, it means the model is not correctly modelling the data. Is there anything instrumental e.g. physical filters that change over at 310 nm that could be the cause of this artefact? Is it still present in the 2 component model?

Line 142. If you performed a single PARAFAC model, then the location of the modelled fluorescence can't change over time. So how can the location of the peak 'move' from LGM to Holocene? Is this from extra PARAFAC analyses that the reader doesn't know about? Or is it a subjective analysis of the original EEMs?

Line 145-150. I would disagree with this interpretation. This fluorescence is typical of 'peak A' and 'peak C' compounds. A peak 'M' fluorescence would be blue shifted compared to 'peak A', and in your component C3 there are two peaks and they both have the same emission wavelengths.

Lines 153-155. The fact that no one else has reported your fluorescence peaks is either very exciting or very worrying. It would suggest that what you are seeing is not anything that has been reported before e.g. you are not seeing 'tyrosine-like' fluorescence, and by implication, you can't definitively interpret it as a microbial signal.

Line 160 and Figure 4. The authors state that Figure 4 shows the 'PARAFAC components', but there is just one line. What is this? Is it C1, or C2, or C3? All three components must be shown individually, here and in Figure 3.

Line 175. C1 and C2 PARAFAC model scores need to be plotted in Figure 3.

Line 177. From Figure 3, component C2 is not 'tryptophan-like'.

Line 184. C3 PARAFAC model scores need to be plotted in Figure 3.

Line 188. There is no evidence in the contemporary literature that fluorescence in this region is recalcitrant.

Line 189-191. This observation is unremarkable, as all humic and fulvic substances standards have a higher fluorescence intensity at the short excitation wavelength (see examples in Aiken (2014)).

Line 191. It sounds like you are saying that there are plants and soil in the ice? I'm sure you don't mean that?

Line 191. The C3 data needs to be presented in Figure 3.

Line 214. No fluorescence data over time is presented (except for the total fluorescence, which is not meaningful). So this section is speculative.

Andy Baker Sydney, Australia

---

## Short Comment (SC2) · 24 Jan 2017

RC1

The reviewer comments are numbered for reference. Each reply is listed below the numbered reviewer comment.

1.Title and text. The term "organic matter quality" seems to be not adequate to describe the measurements here reported. Really, just fluorescence measurements were carried out and interpreted as signatures of some classes of organic components-like markers. I'd suggest the term "organic matter markers" or "organic fluorescent components".

Fluorescence measurements were carried out and interpreted as signatures of organic

components. The chemical nature of the fluorescent fraction of the organic matter was surveyed using a fluorescent technique, thus organic matter markers is an appropriate alternative for the title and text. In the organic matter community, the words/phrases quality, composition, and chemical nature are interchangeably used to infer the same meaning from fluorescent measurements. We aim to define organic matter markers appropriately in the text to clarify any confusion and improve compatibility with both the ice core and organic matter characterization communities.

2. Line 17 and several other points. Usually, time unit is expressed as "kyr" and not as "kyrs". Please, correct in the text and figures.

We will make this adjustment accordingly.

3. Lines 20-22. Here or in the "Results" section, Authors should clarify what They mean with the terms "labile microbial OM", "recalcitrant OM", "bioavailable carbon species" etc. A very short description of these terms could help the reader in better understanding the different biological significance and the different availability in carbon exchange between cryosphere and other ecosystems.

Towards the point mentioned earlier, the descriptions of labile (easily altered) and recalcitrant (less easily altered) descriptions will also help clarify the terms organic matter markers in the manuscript. We will make this edit in the appropriate sections.

4. Line 32. Please, cite also Wolff et al., Southern Ocean sea-ice extent, productivity and iron flux over the past eight glacial cycles. Nature, 2006, Vol. 440, 491-496, doi:10.1038/nature04614.

We will make this adjustment accordingly.

5. Lines 51 and following. What "OM character" means? Chemical composition? Chemical-species or functional groups identification? Authors are requested to clarify their thought.

We will provide descriptions as already mentioned above.

6. Line 55. Even methane formed in anaerobic conditions is a strong forcing factor in the warming climate.

Would the reviewer clarify what is meant by this comment? We describe the release of organic material upon melting of polar ice and the potential for it to be metabolized to carbon dioxide, thus increasing greenhouse concentrations in the environment. Indeed, the anaerobic production of methane is also a strong forcing factor in a warming climate. We have only inferred aerobic production of carbon dioxide in this sentence. Is the reviewer describing the potential for methane to be produced under anaerobic conditions in the ice, and then released as gas?

7. Line 72. Since snow density is variable, it is better to express the mean accumulation rate as cm or mm "water equivalent".

The accumulation rate is provided as cm per year. What is the significance of using "water equivalent"?

8. Line 80. Please, change "drilling solvent" with "drilling fluid".

We will make this adjustment.

9. Line 88. Please change "combusted" with "pre-fired".

The usage of combusted to describe furnace glassware is common in the organic matter community. What is the significance of using "pre-fired" instead?

10. Section 2.4. Even if a reference is cited, Authors are requested to give some basic information about the PARAFAC multivariate analysis.

This section was truncated upon a previous revision. Basic information can be provided. 11. Lines 126 and 128. Authors are requested to shortly describe the characteristics of "bioavailable carbon species" and "more recalcitrant species".

We will describe these phrases in further detail to clarify.

12. Lines 129-131. This early Holocene peak of fluorescent mater is interesting, as well as the larger peak around 21-22 kyr BP. Authors do not discuss these two features in the temporal profile of the WD ice core. I'd like to know the Author interpretation on these large depositions of organic fluorescent compounds, even if as a tentative hypothesis. It should be very interesting to perform some qualitative analysis (e.g., by HPLC-MS measurements) on these samples in order to clarify the nature of the fluorescent compounds.

The fluorescent peaks were discussed as intensity shifts in each climate period, and do not directly correspond with large depositions of organic fluorescent compounds. Rather, the quantum yields of specific fluorescing material is represented, along with the hypotheses that both fluorescing material and concentration of organic material may be contributing to shifts in fluorescence intensities. Large deposition events of organic material cannot be linked to shifts in fluorescence intensities. This point can be clarified further in the text. Regarding the Holocene peak, the authors described a series of years that correspond to that shift in fluorescence, again not related to one event or year. Any anomalies, increases, or even decreases in chemical concentrations, dust, etc. in the WD data set were surveyed to support a tentative hypothesis for this signature, however, none were identified. Further analyses of these samples is unavailable.

13. Lines 132 and following. I surely do not want to minimize the contribution of the PARAFAC analysis, but I have to note that the result of its application is quite basic. From Figure S1, the separation of the fluorescent bands at 420 nm Em and 300 nm Em is very clear even without any multi-parametric analysis. The only significant result is the identification of two fluorescent components C1 and C2 at short Em and Ex wavelength. However, the two components are just attributed to two large organic compound classes (amino acid-like fluorescent compounds), without a more specific characterization. Besides, the C1 and C2 fluorescent components are not clearly differentiated in terms of biological origin: C1 is attributed to tyrosine-like fluorescent

compounds associated to "microbial processing in aquatic environment", while C2 is described as a fluorescent signature overlapping "between tyrosine- and tryptophan-like" fluorescent compounds. At line 177-178, Authors just report that C2 containing tryptophan-like fluorescence could represent "intact dissolved proteins . . ..freshly derived from microorganisms". Authors are requested to better organize, in the present section, the discussion on the possible origin of these components and to enlighten the biological and environmental differences. In conclusion, the PARAFAC analysis seems to be not able to "resolve the representative subset of samples into individual OM fluorescing components", as the Authors assessed at lines 132-133. Even the comparison with the OpenFluor database components did not give significant matches (if I have well understood lines 153-155).

The separation of the fluorescent bands at 420 nm Em and 300 nm Em is very clear even without any multi-parametric analysis. This is correct, however Figure S1 highlights examples of different types of fluorescing organic matter, so it was our intention to show notably obvious differences from the WD core. The WD core fluorescent data set comprised a small fraction of material fluorescing in Figure S1 (b), thus we needed to apply a statistical tool, PARAFAC analysis, to decompose the EEMs into individual fluorescing components, even for fluorescing material at lower Ex/Em wavelength pairs. The significant result is that PARAFAC was used as a multiparametric tool to decompose the EEMs data set into three fluorescing components. That information was subsequently used to identify the chemical character of the fluorescing organic material in each climate period. "Besides, the C1 and C2 fluorescent components are not clearly differentiated in terms of biological origin: C1 is attributed to tyrosine-like fluorescent compounds associated to "microbial processing in aquatic environment", while C2 is described as a fluorescent signature overlapping "between tyrosine- and tryptophan-like" fluorescent compounds." That is the correct interpretation and C1 and C2 fluorescing components cannot be clearly differentiated in terms of biological origin using this fluorescence technique. More specific characterization of C1 and C2 fluorescing components cannot be determined using this bulk analytical technique. This

can be clarified in the text. The chemical species associated with PARAFAC C1 and C2 were discussed in the text providing a bulk representation of the organic materials present throughout 21,000 years. This sets the foundation for future work, a point which can also be clarified in the text. "Authors are requested to better organize, in the present section, the discussion on the possible origin of these components and to enlighten the biological and environmental differences." The discussion on the possible origin based on these data is present in the manuscript, highlighting environmental differences over time. "In conclusion, the PARAFAC analysis seems to be not able to "resolve the representative subset of samples into individual OM fluorescing components", as the Authors assessed at lines 132-133. Even the comparison with the OpenFluor database components did not give significant matches (if I have well understood lines 153-155)." The authors disagree. The PARAFAC analysis resolved the representative subset of samples into the only individual OM fluorescing components that were present in the samples. PARAFAC analysis is capable of producing brilliant results of the data set asked of it. With most EEMs resulting in the example provided in Figure S1, it was not surprising to have the low Ex/Em wavelength fluorescent components modeled as two individual components C1 and C2, prior to C3. The order of the modeled components describes the variation in the data set, and was statistically validated with the drEEM program in MATLAB. The OpenFluor database contains various data sets from samples collected around the world. OpenFluor is not a requirement, and is currently still in its growing phases. Scientists are encouraged to upload their PARAFAC datasets upon publication, but it is not required, thus the database is does not encompass all possible fluorescent component data. OpenFluor matches with the dataset describe PARAFAC components that have been identified in other ecosystems. A match or no match describes unique data worth reporting. We felt it was interesting to report that organic material from 6,000 to 27,000 years ago did not match any of the uploaded PARAFAC data currently in the database. Our dataset is the first of its kind from a continuous Antarctic ice core, thus we stress the importance of its upload to OpenFluor upon publication, which in turn will better serve the fluorescent community.

14. Lines 142-143. The terms "red/blue shifted to longer/shorter Em wavelengths" are repetitions. Please, change in "Em-wavelength red/blue shifted" or "shifted to longer/shorter Em wavelengths". Authors should clarify the statistical significance of these shifts (especially from LGM to LD) and anticipate the consequent biological meaning (especially from LGM-LD to Holocene). Besides, which is the meaning of the red or blue shifts? When blue (red) shift occurs, is the C2 component a marker of tyrosine-like (tryptophan-like) fluorescent compounds?

We can make this adjustment accordingly to clarify the chemical meaning regarding organic matter characterization. A red shift in C2 describes organic matter markers that share similar chemical nature with tryptophan-like species, whereas a blue shift describes more tyrosine-like material.

15. Section 3.2. The relationship between glacial cycles and atmospheric deposition of dust in Antarctica is a very relevant and largely discussed topic in ice core studies. Here, the Authors have to take for granted the inverse relationship between site temperature and dust deposition (by citing the most relevant references) and anticipate the discussion on the possible relationships among temperature, dust and biological activity (or OM transport efficiency), as revealed by the fluorescence temporal profile. At this purpose, Authors should choose the preferred dust indicator among the possible dust markers measured along the WD ice core (nss-Ca, Mn and Sr), also basing on the correlations between the elements (lines 165-166).

The preferred dust indicator of nss-Ca will be highlighted in the discussion section to improve clarity. We will also extend our observations to discuss relationships among temperature, dust, and biological activity relevant to organic matter characterization.

16. Lines 174 and 176. Maybe, "throughout time" is better than "throughout history".

We can make this adjustment accordingly.

17. Line 198-200. Common transport processes of dust and OM could be hypothesized only if dust and OM originated from the same continental areas. In LGM, Southern South America was supposed to be the major dust source area for Antarctica. In LD and, especially, Holocene, even Australia could have played a significant role. Therefore, Authors implicitly suppose that OM was originated in these continental regions. For OM originated by marine sectors (C1, C2?, part of C3), the relationship with dust transport processes cannot be considered significant because they can follow very different pathways (e.g., implying different meridional or zonal atmospheric circulation modes).

We can clarify this point in the text accordingly.

18. Lines 200-201. Authors here refer on relationships between dissolved organic carbon and dust markers. I suppose DOC measurements were not performed as part of this paper (see following sentence in the text). Authors should give more information on that or cite some reference.

DOC concentration measurements were not performed as a part of this paper. Some of our preliminary data include DOC concentrations of the upper firn layers of the WD ice core, which were measured simultaneously with concentrations of nssCa and Sr. We can provide information on how dust and DOC concentrations relate, however do not have this information for the main project.

19. Line 204. I think Authors refer to Figure 4.

Indeed, we did. Thank you. We can make this edit.

20. Lines 205-212. This part has to be completely revised. The complex relationship between dust deposition in Antarctic ice cores and climatic cycles cannot be discussed in this form in this paper and, how I have already pointed out, has been (and will be) the topic for several specific papers. Authors are requested to report the major literature references about LGM-LD-Holocene dust/climate pattern and focus the discussion on the relationship among climate, dust (possibly) and OM fluorescent markers. Besides,

I have to note that the detail in the discussion on the behavior of OM data and dust profiles along the WD ice core is not so high to appreciate specific differences in nss-Ca, Mn and Sr profiles. Therefore, since the three dust-marker profiles were not singularly discussed and differentiated, I'd suggest to replot Figure 4 with just one dust marker (maybe, nss-Ca).

One dust marker will be presented and discussed relevant to the transport of organic matter.

21. Section 4.3. Even this section has to be largely revised. Authors assume a series of speculations to correlate changes of OM fluorescent markers to changes in climatic and environmental conditions, as evaluated by changes in sea-ice coverage (by ss-Na – Authors could add the ss-Na profile in figure 4), dust production and transport (by dust markers) and volcanic eruption frequency (by nss-SO4 spikes) in the LGM, LD and Holocene. However, no reliable comparison among the different time profiles is shown. In particular, while dust and sea ice markers show a progressive decreasing during the LD, the OM fluorescent profile shows an abrupt change (at about 18.5 kyr BP) from high LGM values and very low LD and Holocene levels. All the discussion is too elemental and also the changes in C1 and C2 relative contributions are not clearly interpreted. From the data here reported, I can just see that OM fluorescent markers are high in the LGM, when dust and sea spray are high. However, there is not experimental evidence on which climatic or environmental factors (more efficient meridional or zonal atmospheric transport, larger sea ice coverage, higher input from continental areas, larger emissions from marine biota, etc.) could have driven the OM deposition at the WD site. Finally, the relationship between volcanic activity (as recorded by the nss-SO4 spikes along the WD ice core) and OM fluorescent markers is, in my opinion, really unsustainable. Volcanic signatures in Antarctic ice core are mainly related to long range atmospheric (especially stratospheric) transport of SO2 emitted during eruptions occurred at hemispheric scale and it is really difficult to correlate changes in WD OM to sporadic, short-time and widespread volcanic emissions without a strong experimental

evidence.

Agreed. The co-registered geochemical WD dataset were used to speculate on the origin of the OM characterized by fluorescence spectroscopy. No direct comparisons were reported because none were available for this project; that was beyond the scope of this work. We can clarify this point in the manuscript. Further comments will be organized for the formal response letter.

22. Lines 225-226. What this sentence means? What is compared to the open ocean?

The authors meant to state a comparison of more to less sea-ice extent. We can revise this accordingly.

23. Lines 233-234. Authors are requested to better discuss the red shift of the C2 component, explaining which amino acid-like components increases its contribution to fluorescent OM and at which biological source can be attributed. What "external environments" means?

The red shift clarification will be added as stated above. "External environments" will be revised to clarify that the material originated externally from the englacial ecosystem.

24. Lines 237-243. The pattern of the OM fluorescent markers during the ACR is not visible in Figure 3 (neither in Figure 2). This part is merely speculative and not supported by experimental evidences.

Correct. The dust record was used in Figure 4 as a discussion point to speculate on the variation in organic matter during the ACR, specifically for PARAFAC C2 in the deglaciation.

25. Lines 244-250. How can the Authors explain the very low levels of OM fluorescent markers during the Holocene, when climatic conditions should promote higher terrestrial and marine biological productivity? Which could be the significance of the large spike in OM fluorescent profile (Figure 2) at about 10 kyr BP?

[Figure]

This is a great question, however, it is the intensities that are plotted in Figure 2, thus neither describing high or low levels of OM fluorescent markers, merely just their fluorescent intensities. Fluorescent intensities can be linked to highly or lowly fluorescent material and also chemical concentrations. Without chemical concentrations of OM, we can only speculate to that point. Higher terrestrial and marine biological productivity during the Holocene, as we may assume in the warmest climate for this project, may result in higher fluorescence intensities and different fluorescing OM chemical species in the environment, however, if they are not transported to the WD core, we have no way to detect them with these methods in englacial ice. We cannot discount that carbon productivity is reportedly higher in the Holocene, however, that does not ensure efficient transport of materials to Antarctica. We can thus report our findings and discuss these ideas with the need for future investigations that could answer such questions.

26. Lines 261-262. Authors are requested to clarify how volcanic activity can stimulate OM production. How is calculated the percentage of the fluorescent OM attributed to the volcanic activity? The relationship between volcanic activity and OM deposition at WD site is, in my opinion, not plausible and not supported by experimental data (at least, by experimental data here reported). Have the Authors measured OM fluorescent peaks in ice core sections with volcanic depositions? In absence of experimental support, the discussion about the volcanic activity and OM fluorescent markers should be removed from the manuscript.

Indeed, we do not have experimental support, merely just speculations here.

27. Conclusions section. This part should be changed accordingly to the changes suggested along the different manuscript sections.

This section will be revised accordingly.

All other comments will be addressed in our formal response letter. This response was provided by the lead author based on conversations with a subset of coauthors.

---

## Short Comment (SC3) · 24 Jan 2017

RC2 The reviewer comments are numbered for reference. Each reply is listed below the numbered reviewer comment.

1. You state that "Ice core studies rely on the paradigm that atmospheric deposition is the sole mechanism for specific gases and materials to become trapped in the ice" (Lines 47- 48) yet it is unclear if you apply this paradigm to your work. If you do not allow even a remote chance for in-situ production of this organic matter, then please explicitly state so in your work.

We apply this conservative approach for organic matter preservation in ice cores, however, do so acknowledge the chance for in situ production of organic matter. We can only speculate on the possibility of in situ organic matter production due to method-

ological limitations. We will clarify our approach and acknowledgement of this fact in the text.

2. In lines 179-183 you mention the possibility of in situ OM processing but then do not discuss if such transformation could affect the samples in this work.

A discussion of in situ processing of OM affecting the samples should be added to strengthen this work.

3. You mention that tryptophan-like florescence in C2 may derive from microorganisms, and then mention that the presence of microorganisms may result in in situ OM processing, but step back from linking the two aspects.

We would welcome a section that describes the interwoven nature of microorganisms and organic material, however, with our methodological limitations, were hesitant to include such ideas as no experimental evidence from this project can differentiate labile OM deposited and preserved, or freshly produced material transformed by englacial microorganisms. We are curious to know whether or not that would cause more confusion with this work.

4. In the following paragraph you then mention that Holocene terrestrial plants and soils are the likely source of the C3 OM yet do not mention if in situ processes may affect this material or if you ascribe this material to be solely brought in via atmospheric transport. Please clarify your stance on the source and possible post-depositional processes affecting the samples as both aspects are essential to your interpretations of the data.

Upon clarification of the points mentioned above, this section will be edited accordingly.

5. Please check that all figures are cited in the text. In lines 128-144 you mention Supplemental Figures 1a-b. You do not refer to Figure 2 in the text. As you refer to the Supplemental Figures but not Figure 2, then perhaps their roles should be reversed with the current Figure 2 included in the Supplementary Information and vice versa.

Figure 2 is cited in Line 129. We will double check to make sure all figures are cited accordingly.

6. The left bars and corresponding explanation in the caption of Figure 3 are confusing. In the article text you explicitly state that C3 only occurs during the Holocene. As most readers will likely first look at the figures and captions before reading the article, it bears mentioning in the caption that C3 is specific to the Holocene. Demonstrating the variation in C2 by various time periods (LGM, LD and Holocene) is useful but then makes the reader immediately wonder what is the variation in C1 between climate periods. If there is no substantial variation between time periods for C1, please mention this fact in the caption.

Figure 3 caption will be edited to specify that C3 is specific to the Holocene and that no variation between time periods for C1 was observed. Specifically, the variation in C2 is discussed in the main text, along with the result of no variation for C1, so an addition of that information to the caption will help clarify any confusion for the figure.

7. This sentence is confusing (Lines 227-229): "During the LGM, tundra ecosystems covered more expansive areas of the Earth (Ciais et al., 2012) and while C was cycling, productivity in the environment differed from warmer climates (Ciais et al., 2012 and references within)". Do you mean due to the colder temperatures and increased ice cover and tundra during the LGM, that net C productivity was less than in the other warmer times periods of this study?

Yes, that was the intended meaning of that sentence.

8. The final conclusion overstates the results of the study. To state that labile, microbially derived OM "were the greatest contributors to Earth's atmospheric composition throughout history" is not correct. Labile OM may have been the greatest contributor of total OM in the atmosphere over the time periods covered in this paper, but this situation may not be the case before the LGM. In addition, in this sentence it is not clear what aspect of the "Earth's atmospheric composition" that you mean.

[Figure]

Correct, this sentence is an overstatement, and will be adjusted to reflect that labile OM may have been the greatest contributor of total OM in the atmosphere from 27,000 to 6,000 years ago. We will clarify which aspect of the Earth's atmosphere we are referring to in this section. We stated in the introduction that the significance of ice core findings can produce information about the Earth's lower atmosphere. Would that suffice for this clarification?

9. Line 16 = Define PARAFAC as this is the first time that you use this acronym.

A definition of multivariate parallel factor analysis (PARAFAC) will be edited in this line.

10. Line 48: Place "idea" after "that" in "Extending that to include".

The word "idea" will be included. The omission was an oversight.

11. Line 222: Remove the comma after "LGM".

Indeed, this was a typo, thank you.

This response was provided by the lead author based on conversations with a subset of coauthors.

---

## Short Comment (SC4) · 24 Jan 2017

RC3

The reviewer comments are numbered for reference. Each reply is listed below the numbered reviewer comment.

1. The first weakness of the manuscript is the use of poorly defined wording rendering difficult (sometimes obscure) the reading of the manuscript. For instance, I guess that, when saying "OM quality", you mean "fluorescent signal of the OM"? Also what is a recalcitrant OM?

Fluorescence measurements were carried out and interpreted as signatures of organic components. The chemical nature of the fluorescent fraction of the organic matter was

surveyed using a fluorescent technique, thus organic matter markers is an appropriate alternative for the title and text, as recommended by Reviewer #1. In the organic matter community, the words/phrases quality, composition, and chemical nature are interchangeably used to infer the same meaning from fluorescent measurements. We aim to define the terms used more appropriately upon revision to clarify any confusion and improve compatibility with both the ice core and organic matter characterization communities.

2. Some abbreviates appear in the text without definition. For instance, what is the PARAFAC model that is already mentioned in the abstract, also please indicate what is the basic of this kind of model?

A definition of multivariate parallel factor analysis (PARAFAC) will be edited in the abstract and basic information can be provided about the analysis in the main text.

3. The abbreviates C1, C2 and C3: I guess that they refer to component 1 etc (and not to C1 carbone chain etc).

Correct. The annotation of the abbreviation is set first in Line 135, but can also be annotated in the abstract for clarity.

4. In section 2.3, please define A254 and re-define EEMs here.

A definition of A254 will be provided. EEMs are defined in the Introduction section. Why would another definition be necessary here?

5. Section 2.4: I don't understand the following sentence "A three component PARAFAC model was generated for the subset of samples by drEEM and the N-way toolbox scripts Âż : what is Âń drEEM Âż and N-way ?, please define.

Definitions of each can be provided along with the reference. drEEM is an acronym for a commonly used PARAFAC modelling tool created by Murphy et al. "Decomposition routines for Excitation Emission Matrices" version 0.1.0. The package is compatible for MATALB users, and contains a plethora of MATLAB scripts written to specifically characterize complex OM mixtures worldwide using fluorescence spectroscopy datasets.

6. Concerning units: Line 98 : what is au ? The unit description was not provided. Absorbance units will be annotated to clarify the definition of a.u.

7. I will avoid the use of RU for Raman unit (RU is sometimes used for relative unit). Also I am not sure that the readers of CP, specially those working on ice cores, are familiar with this Raman unit ? A few words on that would help (see also my comment on Figure 2).

Raman Units (R.U.) are the technical unit from the fluorescence instrument and are appropriate for this work. A definition and explanation in a few words will be added upon revision.

8. Introduction, first paragraph (lines 31-446): This paragraph can be improved significantly, for both the wording and the cited references. Two of your co-authors have a nice expertise on the chemistry of ice cores, they certainly can also help here. From my side I would suggest to start with an overall sentence: "In addition to its water stable isotope content that provides a proxy record of past temperature (see Dansgaard et al. (1993), for instance), ice archives atmospheric information on trace gases like $CO_2$ and $CH_4$ encapsulated in air bubbles and chemical species trapped in the ice lattice. Numerous inorganic species trapped in ice has been used to reconstruct past chemical composition of the atmosphere, its recent change in response to growing human activities as well its past natural variability (see Legrand and Mayewski for a review)."

These revisions are greatly appreciated and will be considered at length upon editing the manuscript.

9. I here agree with another reviewer of the manuscript that the Nature paper from Wolff and co-workers (2006) is an excellent example that you have to mention of what was done on deep Antarctic ice cores in terms of changing sea-ice dust emission and marine biological productivity over the 8 climatic cycles. Then focus on what was done

on organics saying "In contrast, as reviewed by Legrand et al. (2013), information on the load and composition of the organic matter archived in ice are still very limited. Âż

Indeed, we will incorporate this into the revised manuscript.

10. I think you can find in this review paper relevant references that can be useful for your introduction. In particular, I suggest to report the work from Amanda Grannas made of the nature of OM in polar ice and those done on the HULIS like content of ice.

Grannas' work involved sampling snow events to test OM photoreactivity, therefore, modern events, but at some point in time, so were the WD ice core samples. Upon revision of the introduction, this may be a good addition to cite as OM in polar ice is still in its infancy.

11. Section 2.2: line 86: what is the difference between cracks and fractures?

This is a good question. Is there a technical definition, or should we just revise this to list one? We can revise this statement to include the difference between a section of ice that contained a break in the continuity, versus a section that was broken all the way through.

12. Section 2.5: Please write a few sentences explaining why your choice was to show these inorganic species. Note that, as far as I know (and checking your fig 4), I see no reason to use three species (Mn, Sr, and Ca) for dust (except if you have in mind to discuss the ratio between the 3 in view to eventually highlight the source region, which seems not to be the case).

The text will be revised to discuss the dust maker of Ca for clarity.

13. Figure 2: Are there any possibility of estimate from the Raman values how much is the concentration of OM ? Indeed, given the scarcity of data on organics, even an order of magnitude would be welcome here. From that and using a typical conversion factor OM/C you can estimate the TOC or DOC content of ice. Also I am surprised that the spikes shown in the fluorescence intensity during the LGM are not more commented

in the text.

Regarding the Holocene peak, the authors described a series of years that correspond to that shift in fluorescence, which are not related to one event or year. Any anomalies, increases, or even decreases in chemical concentrations, dust, etc. in the WD data set were surveyed to support a tentative hypothesis for this signature, however, none were identified. Further analyses of these samples is unavailable.

14. Line 184-195: I assume that "Humic-like fluorescent OM" corresponds to Humic like substances observed in the atmosphere of many regions. If correct, did you consider these species as primary emitted (with soil particles for instance) or secondary produced from oxidation of gaseous organic precursors emitted by the continental biosphere (vegetation)?

Those are good considerations and can be clarified as speculations in the discussion section accordingly. Unfortunately, our methodological limitations prevent us from differentiating the two categories, but an acknowledgement to that point can be added in the text.

All other comments will be addressed in our formal response letter. This response was provided by the lead author based on conversations with a subset of coauthors.

---

## Short Comment (SC5) · 24 Jan 2017

SC1

The short comments are numbered for reference. Each reply is listed below the numbered comment.

1. Line 88. When I have investigated the use of 'septa sealed vials', I find a contaminant fluorescent signal coming from the septa, which in my tests has always been fluorescent. Can the authors confirm that their septa sealed amber glass vials produced zero fluorescence blanks?

We cannot confirm that our septa sealed amber glass vials produced zero fluorescence blanks. We specifically selected septa seals made with Teflon to avoid any carbon and

fluorescent contamination. What type of septa produce fluorescence? What kind of fluorescent signal was detected? If we had fluorescent contaminants originating in our septa, wouldn't that signal be consistent across all our samples?

2. Line 89-90. Following on from my previous comment, were the blanks run just on the melting system, or the melting system and amber glass vials? It is not clear at present.

Blanks were run through the melting system. Blanks were not collected into the discrete sample vials. This will be clarified in the text. Blanks were also run through the melting system into a targeted ultraviolet biological sensor (TUBS) spectrofluorometer, which uses an excitation wavelength of 224nm and collects emission from 280-400nm. All readings of blanks through this unit showed no fluorescence within the 280-400nm emission range, characteristic of dissolved organic material.

3. Line 97. What was the actual absorbance values? These should be plotted as a time series, as A254 is used as a surrogate for DOC in terrestrial systems. It would be interesting for the reader to see this data and for the authors to compare values to other terrestrial systems (e.g. rivers, groundwaters).

All absorbance values were reported below the MQ Water blank run on each day, therefore no values can be used to interrogate the quantity of DOC.

4. Line 106-107. Were the data also processed to remove Rayleigh-Tyndall scatter? How were the Raman and Rayleigh-Tyndall scatter lines processed? Were they replaced by zeros, by NaN (not a number) or was data interpolated? All of these effects can have subtle influence on the resultant PARAFAC model, so it is good to report them.

The EEMS were post-processed to remove the Rayleigh-Tyndall scattering using a MATLAB script of smootheem.m in drEEM version 0.1.0; Murphy et al. 2014. A reference can be added to the text to clarify the smoothing technique to remove each scattering effect.

5. Lines 108-110. The authors must specify what they did for sample classification, normalisation and subset selection. It will be different from Cawley et al (2012), which is just one fluorescence case study, and on pulp mills, so not really very relevant to this research.

We will revise this section to include further details on the procedure for sample classification, normalization, and subset selection prior to PARAFAC modeling. A representative data set was used for PARAFAC modeling, not the entire dataset, so this information will be included to assist others in the same situation.

6. Lines 108-110. Somewhere in this section the authors must quote the value of the standard(s) that they were using. This could be the Raman intensity of Milli-Q water at a specific wavelength, or the intensity of quinine sulphate standards run using the same instrument configuration, or an International Humic Substances Standard, or a tryptophan or tyrosine standard.

The Raman intensity of the MQ Water at a specific wavelength will be provided upon revision. Other standards originally investigated, but not relevant to the manuscript in its current form, were deleted upon a previous revision. We aim to clearly discuss the fluorescent nature of our signatures and will include the standards used for reference in this project. All information will be provided upon revision of the manuscript.

7. Lines 110-111. More detail is needed on the PARAFAC model, to allow the reader to assess its strength in modelling the data. It is crucial in this paper, as the PARAFAC model is the crux of the whole analysis and interpretation. 1. One would expect to see the core consistency value given. A 'passable' model could be considered have a value of >90%, and a good model a score of >99%. 2. It would be very informative to know why the authors chose a 3 component model over a 2 or 4 component model – did the 4 component model try to model noise, for example? Or did it model a plausible 4th component, but with a low core consistency. 3. The percentage of the data fitted by each component is very valuable information, especially if compared with that from

a two and four component model. 4. And finally, a split-half analysis is very useful, especially if the authors perform a split half analysis using randomly split datasets and a split half analysis with LGM data in one dataset and Holocene data in the other. If the split half analysis fails on the latter test, then it tells you that the LGM and Holocene need different PARAFAC models.

1. The core consistency value can be provided. 2. We will explain the rationale for a 3 component model over a 2 or 4 component model. 3. We can report the percentage of the data fitted by each component to strengthen our argument for a 3 component model. 4. Split half analysis was used for this PARAFAC model and will be highlighted in the text upon revision.

8. Line 126-127. Amino-acid like fluorescence is too general. Only tryptophan and tyrosine have aromatic groups which fluoresce, and even then, without independent amino acid analyses to confirm their presence, one can never be sure that these compounds are responsible for the fluorescence. If the fluorescence is from an amino acids source, then C1 and C2 look most like a 'tryrosine-like' compound. Tyrosine would excite at both âĹij225 nm and âĹij275 nm and emit at about 310 nm. But the molecular structure is such that you must observe both the 225 and 275 nm excitation of the 310 nm emission, not just one or the other, as you show in Figure 3. Supplemental Figure 1 confirms the absence of a âĹij275 nm excitation peak. Therefore C1 and C2 are not 'tyrosine-like' or 'tryptophan-like'. Model compounds and contaminants that exhibit a single peak in this general region include simple phenols such as cresol (see Aiken, 2014 in Coble et al. (eds) Aquatic Organic Matter Fluorescence), PAHs such as fluorene (Ferretto et al 2014, DOI:10.1016/j.chemosphere.2013.12.087) and aviation fuel (see Baker et al. 2014, Encyc. Anal. Chem. DOI: 10.1002/9780470027318.a9412).

Similar to the previously mentioned reply, we will incorporate our standard fluorescent references, pertinent concentrations, and the interpretation of our results with such standards, and other references in the literature. We aim to provide a detailed discussion of the possible chemical species responsible for the fluorescence of PARAFAC

components C1-C3.

9. Line 128. The reference to 'recalcitrant species' is speculative. It would be better to specify the excitation and emission wavelengths of this peak or peaks. I am not aware of fluorescence in this region being recalcitrant – instead bio- and photo- degradation studies show that it is degradable (for example, Osburn et al and Stedmon and Cory, both in Coble et al 2014).

We can correct our usage of 'recalcitrant' in this manuscript and cite the appropriate references.

10. Line 129 and Figure 2. There is almost no meaning in 'total OM fluorescence intensities'. Each fluorophore has a different fluorescence efficiency. For example, in this study, you identify three fluorescent components, but each will have a different amount of emitted fluorescence per g C present. So, summing the three is meaningless. It is particularly relevant as low molecular weight compounds such as tryptophan-like and tyrosine-like compounds (argued to be C1 and C2 here) have less chance of their emitted fluorescence being reabsorbed within the molecule, and they therefore have relatively high fluorescence efficiency. In contract, fulvic-like compounds (arguably C3 here) can reabsorb their emitted fluorescence, resulting in a much lower fluorescence efficiency. Figure 2 is therefore just meaningless and instead each PARAFAC component score (C1, C2, C3) needs to be presented.

We will take this information into consideration upon revision of the manuscript. With a subset of samples used for PARAFAC modeling, we did not include fluorescent percentages and intensity tracking because it would not effectively represent the depth (age) profile from sample to sample. Rather, the data points would be spread out unevenly over the depth profile of the ice core. This is an issue we are still discussing to create a way to visualize our fluorescent measurements over time.

11. Line 134 and Figure 3. The PARAFAC scores for C1, C2 and C3 need to be presented in Figure 3. At the moment, no raw data from the PARAFAC model is presented in the paper, yet this is the main focus. The reader has no way of seeing the data and judging its nature e.g. variability over time. Just drawing some PARAFAC model EEMs over an x-y plot would be unacceptable to the fluorescent organic matter research community.

Loading scores can be presented upon revision of the manuscript, however the variability cannot be assessed over time with the subset of samples for this PARAFAC model. Detailed descriptions can be provided outlining the different routes that lead to this specific PARAFAC model, which will be very relevant to the fluorescent organic matter research community.

12. Line 134-136. This observation needs quantification (see comment above).

That information can be provided upon revision.

13. Line 137-139. As in my earlier comment, you cannot have just one of the two excitation peaks that 'tyrosine-like' compounds excite at, and then call it 'tyrosine-like'.

We can address this by discussing the types of chemical species that would fluoresce in that region.

14. Line 139-145. There is most fluorescence at 310 nm, so this is not 'tryptophan-like' at all, as this would also have a peak at 350 nm. More fundamentally, there is a line through the EEM at 310nm which cannot be real. Is this an artefact of the design process of Figure 2, or is it in the actual PARAFAC model? If the latter, it means the model is not correctly modelling the data. Is there anything instrumental e.g. physical filters that change over at 310 nm that could be the cause of this artefact? Is it still present in the 2 component model?

Yes, this feature was present in the 2 component model. As stated previously, more detailed descriptions outlining the success of this particular PARAFAC model will be provided.

15. Line 142. If you performed a single PARAFAC model, then the location of the

modelled fluorescence can't change over time. So how can the location of the peak 'move' from LGM to Holocene? Is this from extra PARAFAC analyses that the reader doesn't know about? Or is it a subjective analysis of the original EEMs?

Extra PARAFAC analyses were performed and can be explained more clearly in the text. It was not a subjective analysis of the original EEMs.

16. Line 145-150. I would disagree with this interpretation. This fluorescence is typical of 'peak A' and 'peak C' compounds. A peak 'M' fluorescence would be blue shifted compared to 'peak A', and in your component C3 there are two peaks and they both have the same emission wavelengths.

This interpretation has shifted through the revision of previous copies of the manuscript. Text clarifying the signatures of A and C peak fluorescence will be revised accordingly.

17. Lines 153-155. The fact that no one else has reported your fluorescence peaks is either very exciting or very worrying. It would suggest that what you are seeing is not anything that has been reported before e.g. you are not seeing 'tyrosine-like' fluorescence, and by implication, you can't definitively interpret it as a microbial signal.

Of the data available in the OpenFluor database, a repository of a selection of samples, and not every fluorescent study completed, our results showed no matches with other PARAFAC components. This is reasonable given the scope of the project and the great volume of samples spanning 6,000 to 27,000 years ago from ice. Yes, we agree that what we are seeing is not anything that has been reported before. We also agree that your suggestion as the correct interpretation of the PARAFAC components would not distinctly be tyrosine-like or tryptophan-like, thus the interpretation of a microbial signal is not definitive. These interpretations will be edited accordingly upon revision.

18. Line 160 and Figure 4. The authors state that Figure 4 shows the 'PARAFAC components', but there is just one line. What is this? Is it C1, or C2, or C3? All three components must be shown individually, here and in Figure 3.

The PARAFAC components determined in each climate period are provided on the same graph for your convenience. The black line refers to the $\delta$18O record. This can be clarified in the text and caption.

19, Line 175. C1 and C2 PARAFAC model scores need to be plotted in Figure 3. Line 184. C3 PARAFAC model scores need to be plotted in Figure 3.

Model scores can be provided.

20. Line 189-191. This observation is unremarkable, as all humic and fulvic substances standards have a higher fluorescence intensity at the short excitation wavelength (see examples in Aiken (2014)).

Correct. We can adjust this appropriately and provide the reference.

21. Line 191. It sounds like you are saying that there are plants and soil in the ice? I'm sure you don't mean that?

Yes, thank you. This was an error in phrasing and can be corrected upon revision.

22. Line 214. No fluorescence data over time is presented (except for the total fluorescence, which is not meaningful). So this section is speculative.

See previous comment regarding our PARAFAC model and the nature of the fluorescent representative dataset. More details will be provided outlining the nature of this model.

This response was provided by the lead author based on conversations with a subset of coauthors.

---

## Author Comment (AC1) · 9 Feb 2017

RC1

The reviewer comments are numbered for reference. Each reply is listed below the numbered reviewer comment.

1. Title and text. The term "organic matter quality" seems to be not adequate to describe the measurements here reported. Really, just fluorescence measurements were carried out and interpreted as signatures of some classes of organic components-like markers. I'd suggest the term "organic matter markers" or "organic fluorescent components".

Fluorescence measurements were carried out and interpreted as signatures of organic

components. The chemical nature of the fluorescent fraction of the organic matter was surveyed using a fluorescent technique, thus organic matter markers is an appropriate alternative for the title and text. In the organic matter community, the words/phrases quality, composition, and chemical nature are interchangeably used to infer the same meaning from fluorescent measurements. We aim to define organic matter markers appropriately in the text to clarify any confusion and improve compatibility with both the ice core and organic matter characterization communities. Proposed title revision, "A 21,000 year record of organic matter markers in the WAIS Divide ice core."

2. Line 17 and several other points. Usually, time unit is expressed as "kyr" and not as "kyrs". Please, correct in the text and figures.

All time units listed as "kyrs" will be adjusted to "kyr" in the text and figures.

3. Lines 20-22. Here or in the "Results" section, Authors should clarify what They mean with the terms "labile microbial OM", "recalcitrant OM", "bioavailable carbon species" etc. A very short description of these terms could help the reader in better understanding the different biological significance and the different availability in carbon exchange between cryosphere and other ecosystems.

Towards the point mentioned earlier, these descriptions will also help clarify the terms organic matter markers in the manuscript. Labile OM/bioavailable carbon species are both defined as the fluorescent fraction considered to be easily altered by microorganisms, whereas recalcitrant OM refers to the chemical species that are less easily altered in the environment. We will make these edits in the appropriate sections.

4. Line 32. Please, cite also Wolff et al., Southern Ocean sea-ice extent, productivity and iron flux over the past eight glacial cycles. Nature, 2006, Vol. 440, 491-496, doi:10.1038/nature04614.

As per this suggestion, we will make this adjustment accordingly.

5. Lines 51 and following. What "OM character" means? Chemical composition?

Chemical-species or functional groups identification? Authors are requested to clarify their thought.

As mentioned above, we will provide appropriate definitions and descriptions. Chemical composition will be discussed in the manuscript as the fluorescent nature of the OM, along with specific details corresponding to higher/lower molecular weights, aromaticity, reactivity, and potential functional groups identified.

6. Line 55. Even methane formed in anaerobic conditions is a strong forcing factor in the warming climate.

Could the reviewer clarify what is meant by this comment? We describe the release of organic material upon melting of polar ice and the potential for it to be metabolized to carbon dioxide, thus increasing greenhouse concentrations in the environment. Indeed, the anaerobic production of methane is also a strong forcing factor in a warming climate. We have only inferred aerobic production of carbon dioxide in this sentence. Is the reviewer describing the potential for methane to be produced under anaerobic conditions in the ice, and then released as gas?

7. Line 72. Since snow density is variable, it is better to express the mean accumulation rate as cm or mm "water equivalent".

We will change this to average annual accumulation rate 0.207mweq a-1 (Banta et al., 2008).

8. Lines 74-76. What means this sentence? Several other ice cores (for instance, Taylor Dome and Talos Dome, in the same Antarctic Sector; Dome C and Dome Fuji, in the inner Antarctica; Dronning Maud Land, in the Atlantic Sector; etc.), even drilled before WD ice core, constitute "equivalent paleoclimate record" to Greenland ice cores. In particular, the EDC, EDML and DF climate records were compared with the climate oscillations recorded along the NGRIP ice core in: EPICA Community Members, Oneto-one coupling of glacial climate variability in Greenland and Antarctica. Nature,

2006, Vol. 444, 195-198, doi:10.1038/nature05301.

We acknowledge that other ice cores were investigated in Antarctica to be equivalent paleoclimate records to Greenland ice cores. We will edit the text to clarify this point, providing appropriate references, and delete the claim previously stated.

9. Line 80. Please, change "drilling solvent" with "drilling fluid".

"Drilling solvent" will be edited to "drilling fluid" in the text.

10. Line 88. Please change "combusted" with "pre-fired".

The usage of the word "combusted" to describe furnaced glassware is common in the organic matter community. "Pre-fired" is a suitable alternative and can be edited in the text.

11. Section 2.3. The correction for the absorbance measurements seems to be not clear. Authors are asked to give more information on that. Besides, the absorbance threshold seems to be quite high. If a.u. means, as I think, absorbance unit, the value A = 0.3 corresponds to a percentage transmittance of 50% (A = Log 1/T) that seems to be too low for ice-core melted water at 254 nm. Maybe, some particles were suspended or some gas bubbles were present in the melted samples during the measurements. Authors are requested to clarify this point.

We can report that all of our samples were optically dilute.

12. Line 97. Maybe the term "optically dilute" could be changed in "optically transparent" (but I do not think that this term is correct for T% = 50%).

The term "optically transparent" can be edited in the text.

13. Section 2.4. Even if a reference is cited, Authors are requested to give some basic information about the PARAFAC multivariate analysis.

Basic information on the setup of our analyses will be added to the manuscript.

[Figure]

14. Section 2.5. Authors should here anticipate why some elements were considered in this paper (e.g., nssCa as crustal marker, ssNa as sea spray indicator, nss-SO4 spikes to identify volcanic deposition signatures, etc.). Besides, more detail is requested in calculating the ss- and nss- fractions of Na, Ca and SO4. Since both Na and Ca can be related to two main sources (sea spray and dust), a four-equation system is necessary to calculate the ss- and nss- fractions (particular attention has to be put in evaluating ssNa during the LGM and nss-Ca during Holocene). Finally, which sea water ratio was used for the calculation of ss-SO4? Have the Authors used the SO4/Na seawater ratio of 0.25 or a lower value?

The crustal marker nssCa will be highlighted in the manuscript instead of three crustal indicators, and appropriate references will be provided to strengthen why some elements were considered for this work. The sea spray indicator (ssNa) and calculation information are referenced already in the text: Bowen 1979, WAIS Divide Project Members 2013.

15. Lines 126 and 128. Authors are requested to shortly describe the characteristics of "bioavailable carbon species" and "more recalcitrant species".

Similar to the third comment above, these descriptions will strengthen the scope of the manuscript and the usage of "organic matter markers" in the text. Labile OM/bioavailable carbon species are both defined as the fluorescent fraction considered to be easily altered by microorganisms, whereas recalcitrant OM refers to the chemical species that are less easily altered in the environment. We will make these edits in the appropriate sections.

16. Lines 129-131. This early Holocene peak of fluorescent mater is interesting, as well as the larger peak around 21-22 kyr BP. Authors do not discuss these two features in the temporal profile of the WD ice core. I'd like to know the Author interpretation on these large depositions of organic fluorescent compounds, even if as a tentative hypothesis. It should be very interesting to perform some qualitative analysis (e.g.,

by HPLC-MS measurements) on these samples in order to clarify the nature of the fluorescent compounds.

The fluorescent peaks were discussed as intensity shifts in each climate period, and do not directly correspond with large depositions of organic fluorescent compounds. Rather, the quantum yields of specific fluorescing material is represented, along with the hypotheses that both fluorescing material and concentration of organic material may be contributing to shifts in fluorescence intensities. Large deposition events of organic material cannot be linked to shifts in fluorescence intensities. This point can be clarified further in the text. Regarding the Holocene peak, the authors described a series of years that correspond to that shift in fluorescence, again not related to one event or year. Any anomalies, increases, or even decreases in chemical concentrations, dust, etc. in the WD data set were surveyed to support a tentative hypothesis for this signature, however, none were identified. Further analyses of these samples is not possible as only 7.5mL of each sample were available and have been used in the present analyses. It is our intention to remove this figure upon revision.

17. Lines 132 and following. I surely do not want to minimize the contribution of the PARAFAC analysis, but I have to note that the result of its application is quite basic. From Figure S1, the separation of the fluorescent bands at 420 nm Em and 300 nm Em is very clear even without any multi-parametric analysis. The only significant result is the identification of two fluorescent components C1 and C2 at short Em and Ex wavelength. However, the two components are just attributed to two large organic compound classes (amino acid-like fluorescent compounds), without a more specific characterization. Besides, the C1 and C2 fluorescent components are not clearly differentiated in terms of biological origin: C1 is attributed to tyrosine-like fluorescent compounds associated to "microbial processing in aquatic environment", while C2 is described as a fluorescent signature overlapping "between tyrosine- and tryptophan-like" fluorescent compounds. At line 177-178, Authors just report that C2 containing tryptophan-like fluorescence could represent "intact dissolved proteins . . ..freshly derived from microorganisms". Authors are requested to better organize, in the present section, the discussion on the possible origin of these components and to enlighten the biological and environmental differences. In conclusion, the PARAFAC analysis seems to be not able to "resolve the representative subset of samples into individual OM fluorescing components", as the Authors assessed at lines 132-133. Even the comparison with the OpenFluor database components did not give significant matches (if I have well understood lines 153-155).

The separation of the fluorescent bands at 420 nm Em and 300 nm Em is very clear even without any multi-parametric analysis. This is correct, however Figure S1 highlights examples of different types of fluorescing organic matter, so it was our intention to show notably obvious differences from the WD core. The WD core fluorescent data set comprised a small fraction of material fluorescing in Figure S1 (b), thus we needed to apply a statistical tool, PARAFAC analysis, to decompose the EEMs into individual fluorescing components, even for fluorescing material at lower Ex/Em wavelength pairs. The significant result is that PARAFAC was used as a multiparametric tool to decompose the EEMs data set into three fluorescing components. That information was subsequently categorized to identify the chemical character of the fluorescing organic material characteristic of each climate period. RC1 Comment: "Besides, the C1 and C2 fluorescent components are not clearly differentiated in terms of biological origin: C1 is attributed to tyrosine-like fluorescent compounds associated to "microbial processing in aquatic environment", while C2 is described as a fluorescent signature overlapping "between tyrosine- and tryptophan-like" fluorescent compounds." C1 and C2 fluorescing components cannot be clearly differentiated in terms of biological origin using this fluorescence technique. More specific characterization of C1 and C2 fluorescing components cannot be determined using this bulk analytical technique, however possible chemical species can be suggested. This can be clarified in the text. The chemical species associated with PARAFAC C1 and C2 were discussed in the text providing a bulk representation of the organic materials present throughout 21,000 years. This sets the foundation for future work, a point which can also be clarified in the text. RC1

Comment: "Authors are requested to better organize, in the present section, the discussion on the possible origin of these components and to enlighten the biological and environmental differences." The discussion on the possible origin based on these data is present in the manuscript, highlighting environmental differences over time. RC1 Comment: "In conclusion, the PARAFAC analysis seems to be not able to "resolve the representative subset of samples into individual OM fluorescing components", as the Authors assessed at lines 132-133. Even the comparison with the OpenFluor database components did not give significant matches (if I have well understood lines 153-155)." The authors disagree. The PARAFAC analysis resolved the representative subset of samples into the only individual OM fluorescing components that were present in the samples. PARAFAC analysis is capable of producing brilliant results of the data set asked of it. With most EEMs resulting in the example provided in Figure S1, it was not surprising to have the low Ex/Em wavelength fluorescent components modeled as two individual components C1 and C2, prior to C3. The order of the modeled components describes the variation in the data set, and was statistically validated with the drEEM program in MATLAB. The OpenFluor database contains various data sets from samples collected around the world. Submission of data to OpenFluor is not a requirement, and is currently still in its growing phases. Scientists are encouraged to upload their PARAFAC datasets upon publication, but it is not required, thus the database it contains does not encompass all possible fluorescent component data. OpenFluor matches with the dataset describe PARAFAC components that have been identified in other ecosystems. A match or no match describes unique data worth reporting. We felt it was interesting to report that organic material from 6,000 to 27,000 years ago did not match any of the uploaded PARAFAC data currently in the database. Our dataset is the first of its kind from a continuous Antarctic ice core, thus we stress the importance of its upload to OpenFluor upon publication, which in turn will better serve the fluorescent community.

18. Lines 142-143. The terms "red/blue shifted to longer/shorter Em wavelengths" are repetitions. Please, change in "Em-wavelength red/blue shifted" or "shifted to

longer/shorter Em wavelengths". Authors should clarify the statistical significance of these shifts (especially from LGM to LD) and anticipate the consequent biological meaning (especially from LGM-LD to Holocene). Besides, which is the meaning of the red or blue shifts? When blue (red) shift occurs, is the C2 component a marker of tyrosine-like (tryptophan-like) fluorescent compounds?

We can make this adjustment accordingly to clarify the chemical meaning regarding organic matter characterization. A red shift in C2 describes organic matter markers that have longer emission wavelengths, therefore correspond to higher molecular weight chemical species that are potentially more aromatic than materials that fluoresce at shorter wavelengths.

19. Section 3.2. The relationship between glacial cycles and atmospheric deposition of dust in Antarctica is a very relevant and largely discussed topic in ice core studies. Here, the Authors have to take for granted the inverse relationship between site temperature and dust deposition (by citing the most relevant references) and anticipate the discussion on the possible relationships among temperature, dust and biological activity (or OM transport efficiency), as revealed by the fluorescence temporal profile. At this purpose, Authors should choose the preferred dust indicator among the possible dust markers measured along the WD ice core (nss-Ca, Mn and Sr), also basing on the correlations between the elements (lines 165-166).

Relationships between dust markers and temperature will be further explained regarding OM transport efficiency. Without concentrations of OM investigated for this project, we were only able to speculate on the relationships between dust concentrations and OM transport efficiency, since we cannot directly relate higher concentrations of dust to higher concentrations of OM. Thus, the OM character reported in this project was discussed in terms of the influence of dust concentrations for the different climate periods. We highlighted that different types of OM were observed for higher and lower concentrations of dust. Three dust markers were presented in this project so that the reader may see different indicators of continental influences. Trends among all three

can be discussed in further detail upon revision, or the records of Mn and Sr can be removed so that our results focus on the influence of the nssCa dust indicator with OM character (e.g., chemical species and characteristics).

20. Lines 174 and 176. Maybe, "throughout time" is better than "throughout history".

We can make this adjustment accordingly.

21. Line 198-200. Common transport processes of dust and OM could be hypothesized only if dust and OM originated from the same continental areas. In LGM, Southern South America was supposed to be the major dust source area for Antarctica. In LD and, especially, Holocene, even Australia could have played a significant role. Therefore, Authors implicitly suppose that OM was originated in these continental regions. For OM originated by marine sectors (C1, C2?, part of C3), the relationship with dust transport processes cannot be considered significant because they can follow very different pathways (e.g., implying different meridional or zonal atmospheric circulation modes).

Common transport processes of OM were only hypothesized, and we inferred a local South American major dust source region for Antarctica. We cannot predict the origin of our OM, but rather make hypotheses and suggestions based on our dataset. We can clarify that point for dust and continental OM from the South American southern region, and separately discuss marine origins having different pathways.

22. Lines 200-201. Authors here refer on relationships between dissolved organic carbon and dust markers. I suppose DOC measurements were not performed as part of this paper (see following sentence in the text). Authors should give more information on that or cite some reference.

DOC concentrations were not performed as part of this work. Correlation values can be presented from the preliminary data that is unpublished. Preliminary data was provided in the Supplemental section and will be referenced accordingly.

23. Line 204. I think Authors refer to Figure 4.

Indeed, we did. Thank you. We can make this edit.

24. Lines 205-212. This part has to be completely revised. The complex relationship between dust deposition in Antarctic ice cores and climatic cycles cannot be discussed in this form in this paper and, how I have already pointed out, has been (and will be) the topic for several specific papers. Authors are requested to report the major literature references about LGM-LD-Holocene dust/climate pattern and focus the discussion on the relationship among climate, dust (possibly) and OM fluorescent markers. Besides, I have to note that the detail in the discussion on the behavior of OM data and dust profiles along the WD ice core is not so high to appreciate specific differences in nss-Ca, Mn and Sr profiles. Therefore, since the three dust-marker profiles were not singularly discussed and differentiated, I'd suggest to replot Figure 4 with just one dust marker (maybe, nss-Ca).

One dust marker (nssCa) will be presented and only discussed in terms of the fluctuations of dust concentrations with the different OM chemical species present in different climate periods. This section will be completely revised to reflect these points.

25. Section 4.3. Even this section has to be largely revised. Authors assume a series of speculations to correlate changes of OM fluorescent markers to changes in climatic and environmental conditions, as evaluated by changes in sea-ice coverage (by ss-Na – Authors could add the ss-Na profile in figure 4), dust production and transport (by dust markers) and volcanic eruption frequency (by nss-SO4 spikes) in the LGM, LD and Holocene. However, no reliable comparison among the different time profiles is shown. In particular, while dust and sea ice markers show a progressive decreasing during the LD, the OM fluorescent profile shows an abrupt change (at about 18.5 kyr BP) from high LGM values and very low LD and Holocene levels. All the discussion is too elemental and also the changes in C1 and C2 relative contributions are not clearly interpreted. From the data here reported, I can just see that OM fluorescent markers are high

in the LGM, when dust and sea spray are high. However, there is not experimental evidence on which climatic or environmental factors (more efficient meridional or zonal atmospheric transport, larger sea ice coverage, higher input from continental areas, larger emissions from marine biota, etc.) could have driven the OM deposition at the WD site. Finally, the relationship between volcanic activity (as recorded by the nss-SO4 spikes along the WD ice core) and OM fluorescent markers is, in my opinion, really unsustainable. Volcanic signatures in Antarctic ice core are mainly related to long range atmospheric (especially stratospheric) transport of SO2 emitted during eruptions occurred at hemispheric scale and it is really difficult to correlate changes in WD OM to sporadic, short-time and widespread volcanic emissions without a strong experimental evidence.

The authors agree. The co-registered geochemical WD dataset were used to speculate on the origin of the OM characterized by fluorescence spectroscopy. No direct comparisons were reported because none were available for this project; that was beyond the scope of this work. We can clarify this point in the manuscript. The PARAFAC components C1 and C2 relative contributions can be discussed in terms of percentages relative to the other components. See above responses outlining our PARAFAC component discussion section in further detail. The sections outlining the volcanic signatures is highly speculative and will be removed upon revision.

26. Lines 225-226. What this sentence means? What is compared to the open ocean?

The authors meant to state a comparison of more to less sea-ice extent. We can revise this accordingly.

27. Lines 233-234. Authors are requested to better discuss the red shift of the C2 component, explaining which amino acid-like components increases its contribution to fluorescent OM and at which biological source can be attributed. What "external environments" means?

The red shift clarification will be added as stated above. "External environments" will be

revised to clarify that the material originated externally from the englacial ecosystem.

28. Lines 237-243. The pattern of the OM fluorescent markers during the ACR is not visible in Figure 3 (neither in Figure 2). This part is merely speculative and not supported by experimental evidences.

Correct. The dust record was used in Figure 4 as a discussion point to speculate on the variation in OM character during the ACR, specifically for PARAFAC C2 in the deglaciation period. We can revise this section to clarify our speculation.

29. Lines 244-250. How can the Authors explain the very low levels of OM fluorescent markers during the Holocene, when climatic conditions should promote higher terrestrial and marine biological productivity? Which could be the significance of the large spike in OM fluorescent profile (Figure 2) at about 10 kyr BP?

This is a great question, however, it is the intensities that are plotted in Figure 2, thus neither describing high or low levels of OM fluorescent markers, merely just their fluorescent intensities. Fluorescent intensities can be linked to highly or lowly fluorescent material and also chemical concentrations. Without chemical concentrations of OM, we can only speculate to that point. Higher terrestrial and marine biological productivity during the Holocene, as we may assume in the warmest climate for this project, may result in higher fluorescence intensities and different fluorescing OM chemical species in the environment, however, if they are not transported to the WD core, we have no way to detect them with these methods in englacial ice. We cannot discount that carbon productivity is reportedly higher in the Holocene, however, that does not ensure efficient transport of materials to Antarctica. We can thus report our findings and discuss these ideas with the need for future investigations that could answer such questions.

30. Line 258. Please, change "Concentrations of nss-sulfur. . ." with "Spikes in nss-SO4 concentrations. . ."

The sections outlining volcanic activity discussed in terms of OM character will be removed.

31. Lines 261-262. Authors are requested to clarify how volcanic activity can stimulate OM production. How is calculated the percentage of the fluorescent OM attributed to the volcanic activity? The relationship between volcanic activity and OM deposition at WD site is, in my opinion, not plausible and not supported by experimental data (at least, by experimental data here reported). Have the Authors measured OM fluorescent peaks in ice core sections with volcanic depositions? In absence of experimental support, the discussion about the volcanic activity and OM fluorescent markers should be removed from the manuscript.

Indeed, we do not have experimental support, merely just speculations on this topic. Volcanic activity discussion sections will be removed accordingly.

32. Conclusions section. This part should be changed accordingly to the changes suggested along the different manuscript sections.

This section will be revised accordingly based on all the reviewer's comments.

A revised Figure 4 is provided for consideration.

―――――――――――――――――

[Figure]

Figure 4. Trace element concentration of (top) non-sea salt calcium (nssCa; ppb), and the δ18O
(per mil) temperature record (WAIS Divide Project Members, 2013) from the West Antarctic Ice
Sheet Divide ice core as a function of time (kyr before present 1950), dating from the Last
Glacial Maximum (LGM), through the last deglaciation (LD), to the mid-Holocene.

References

WAIS Divide Project Members (2013) Onset of deglacial warming in West Antarctica driven by
    local orbital forcing. *Nature* 500, 440-444.
**Fig. 1.** Figure 4. Trace element concentration of (top) non-sea salt calcium (nssCa; ppb), and
the δ18O (per mil) temperature record (WAIS Divide Project Members, 2013) from the West
Antarctic Ice Sheet Divide

---

## Author Comment (AC3) · 9 Feb 2017

RC3 The reviewer comments are numbered for reference. Each reply is listed below the numbered reviewer comment.

1. The first weakness of the manuscript is the use of poorly defined wording rendering difficult (sometimes obscure) the reading of the manuscript. For instance, I guess that, when saying "OM quality", you mean "fluorescent signal of the OM"? Also what is a recalcitrant OM?

Fluorescence measurements were carried out and interpreted as signatures of organic components. The chemical nature of the fluorescent fraction of the organic matter was surveyed using a fluorescent technique, thus organic matter markers is an appropriate alternative for the title and text, as recommended by Reviewer #1. In the organic

matter community, the words/phrases quality, composition, and chemical nature are interchangeably used to infer the same meaning from fluorescent measurements. We aim to define the terms used more appropriately upon revision to clarify any confusion and improve compatibility with both the ice core and organic matter characterization communities.

2. Some abbreviates appear in the text without definition. For instance, what is the PARAFAC model that is already mentioned in the abstract, also please indicate what is the basic of this kind of model?

A definition of multivariate parallel factor (PARAFAC) analysis will be edited in the abstract and basic information can be provided about the analysis in the main text.

3. The abbreviates C1, C2 and C3: I guess that they refer to component 1 etc (and not to C1 carbone chain etc).

Correct. The annotation of the abbreviation is set first in Line 135, but can also be annotated in the abstract for clarity.

4. In section 2.3, please define A254 and re-define EEMs here.

A definition of A254 (absorbance at 254nm) will be provided. The acronym "EEMs" is defined in the Introduction section.

5. Section 2.4: I don't understand the following sentence "A three component PARAFAC model was generated for the subset of samples by drEEM and the N-way toolbox scripts Âż : what is Âń drEEM Âż and N-way ?, please define.

Definitions of each can be provided along with the currently listed reference. "drEEM" is an acronym for a commonly used PARAFAC modelling tool created by Murphy et al. "Decomposition routines for Excitation Emission Matrices" version 0.1.0. The package is compatible for MATALB users, and contains a plethora of MATLAB scripts written to specifically characterize complex OM mixtures worldwide using fluorescence spectroscopy datasets.

6. Concerning units: Line 98 : what is au ?

The unit description was not provided. Absorbance units will be annotated to clarify the definition of a.u. in the text.

7. I will avoid the use of RU for Raman unit (RU is sometimes used for relative unit). Also I am not sure that the readers of CP, specially those working on ice cores, are familiar with this Raman unit ? A few words on that would help (see also my comment on Figure 2).

Raman Units (R.U.) are the technical unit from the fluorescence instrument and are appropriate for this work. A definition and explanation in a few words will be added upon revision.

8. Introduction, first paragraph (lines 31-446): This paragraph can be improved significantly, for both the wording and the cited references. Two of your co-authors have a nice expertise on the chemistry of ice cores, they certainly can also help here. From my side I would suggest to start with an overall sentence: "In addition to its water stable isotope content that provides a proxy record of past temperature (see Dansgaard et al. (1993), for instance), ice archives atmospheric information on trace gases like $CO_2$ and $CH_4$ encapsulated in air bubbles and chemical species trapped in the ice lattice. Numerous inorganic species trapped in ice has been used to reconstruct past chemical composition of the atmosphere, its recent change in response to growing human activities as well its past natural variability (see Legrand and Mayewski for a review)."

We appreciate the suggested wording changes and will update the text accordingly.

9. I here agree with another reviewer of the manuscript that the Nature paper from Wolff and co-workers (2006) is an excellent example that you have to mention of what was done on deep Antarctic ice cores in terms of changing sea-ice dust emission and marine biological productivity over the 8 climatic cycles. Then focus on what was done on organics saying "In contrast, as reviewed by Legrand et al. (2013), information on

the load and composition of the organic matter archived in ice are still very limited."

Indeed, we will incorporate the reference and the suggested phrasing into the revised manuscript.

10. I think you can find in this review paper relevant references that can be useful for your introduction. In particular, I suggest to report the work from Amanda Grannas made of the nature of OM in polar ice and those done on the HULIS like content of ice.

Grannas' work involved sampling snow events to test OM photoreactivity, therefore, modern events, but at some point in time, so were the WD ice core samples. Upon revision of the introduction, this may be a good addition to cite as OM in polar ice is still in its infancy.

11. Section 2.1.: line 75: WD is not at all the first Antarctic ice record available for comparison with Greenland records. Please modify the text.

The text will be modified upon revision.

12. Section 2.2: line 86: what is the difference between cracks and fractures?

This is a good question. We can revise this statement to include the difference between a section of ice that contained a break in the continuity (crack), versus a section that was broken all the way through (fracture).

13. Section 2.5: Please write a few sentences explaining why your choice was to show these inorganic species. Note that, as far as I know (and checking your fig 4), I see no reason to use three species (Mn, Sr, and Ca) for dust (except if you have in mind to discuss the ratio between the 3 in view to eventually highlight the source region, which seems not to be the case).

We will revise the manuscript to report nssCa as our dust marker for this work.

14. Figure 2: Are there any possibility of estimate from the Raman values how much is the concentration of OM? Indeed, given the scarcity of data on organics, even an order

of magnitude would be welcome here. From that and using a typical conversion factor OM/C you can estimate the TOC or DOC content of ice. Also I am surprised that the spikes shown in the fluorescence intensity during the LGM are not more commented in the text.

Regarding the Holocene peak, the authors described a series of years that correspond to that shift in fluorescence, which are not related to one event or year. Any anomalies, increases, or even decreases in chemical concentrations, dust, etc. in the WD data set were surveyed to support a tentative hypothesis for this signature, however, none were identified. Further analyses of these samples is not possible. Upon revision of the manuscript, this Figure will be removed and an updated figure for the PARAFAC model and variation will be provided.

15. Line 184-195: I assume that "Humic-like fluorescent OM" corresponds to Humic like substances observed in the atmosphere of many regions. If correct, did you consider these species as primary emitted (with soil particles for instance) or secondary produced from oxidation of gaseous organic precursors emitted by the continental biosphere (vegetation)?

Those are good considerations and can be clarified as speculations in the discussion section accordingly. Unfortunately, our methodological limitations prevent us from differentiating the two categories, but an acknowledgement to that point can be added in the text.

16. Section 4.2: Your discussion on change of dust tracers is quite oversimplified and I would recommend you to revisit previous works done on this topic.

This section will undergo considerable revisions to highlight the usage of the dust tracers to supplement the results of the WD OM characterization work by fluorescence spectroscopy. We aim to use this information to better understand the different environmental changes over the three climate periods.

17. Lines 255-265: This discussion is from my point of view rather confusing. It is incorrect to say that nssS concentrations are used to trace back volcanic eruptions. Only the narrow peaks of nssS are related to volcanic eruptions whereas the background nssS level in Antarctica originates in marine biogenic emissions (please revisit here the paper from Wolff et al., 2006 for instance). Also, I don't think that the wording of the following sentence makes sense "Therefore, volcanic eruptions increase the potential for particles and chemicals to be transported to polar regions and deposited onto ice-sheets." Please modify.

This section was purely speculative and without any experimental results connecting the OM character with volcanic activity, this section will be removed from the manuscript upon further revisions.

18. Supplementary material: Following your line 201 on a correlation between DOC and nssCa, I checked the S2 figure (extracted below) that strongly bothers me. Indeed, if the DOC unit you report is correct, DOC levels of this Antarctic ice are as high as 200 $\mu$M. If I am right that means 12*200 $\mu$g L-1 i.e. 2400 ppbC. If correct, please comment with respect to the review of Legrand et al. (CP, 2013). It is very likely that you have a large DOC contamination in this shallow WD core. Also, sorry but I don't see a good correlation in this figure between dust and DOC!!! Please comment.

Correct. The shallow WD core DOC concentration numbers are a cause for concern. We cannot validate how this data could have resulted from contamination or is a reflection of surface DOC concentrations and processes. Therefore, we will remove this information from the Supplemental Information section. Upon revising the manuscript, using only nssCa for a discussion on the dust concentration changes with climate, these Supplemental figures are no longer relevant. A discussion linking Ca and Sr to DOC concentration will no longer be a focal point of the manuscript, therefore this material will not be used.

A new Figure 4 is provided for consideration.

[Figure]

Figure 4. Trace element concentration of (top) non-sea salt calcium (nssCa; ppb), and the δ18O (per mil) temperature record (WAIS Divide Project Members, 2013) from the West Antarctic Ice Sheet Divide ice core as a function of time (kyr before present 1950), dating from the Last Glacial Maximum (LGM), through the last deglaciation (LD), to the mid-Holocene.

References

WAIS Divide Project Members (2013) Onset of deglacial warming in West Antarctica driven by local orbital forcing. *Nature* 500, 440-444.

**Fig. 1.** Figure 4. Trace element concentration of (top) non-sea salt calcium (nssCa; ppb), and the $\delta$18O (per mil) temperature record (WAIS Divide Project Members, 2013) from the West Antarctic Ice Sheet Divide

---

## Author Comment (AC4) · 9 Feb 2017

SC1

The short comments are numbered for reference. Each reply is listed below the numbered comment.

1. Line 88. When I have investigated the use of 'septa sealed vials', I find a contaminant fluorescent signal coming from the septa, which in my tests has always been fluorescent. Can the authors confirm that their septa sealed amber glass vials produced zero fluorescence blanks?

We cannot confirm that our septa sealed amber glass vials produced zero fluorescence blanks. We specifically selected septa seals made with Teflon to avoid any carbon and

fluorescent contamination. What type of septa produce fluorescence? What kind of fluorescent signal was detected? If we had fluorescent contaminants originating in our septa, wouldn't that signal be consistent across all our samples? We will edit the text to clarify Teflon septa sealed amber glass vials.

2. Line 89-90. Following on from my previous comment, were the blanks run just on the melting system, or the melting system and amber glass vials? It is not clear at present.

Blanks were run through the melting system. Blanks were not collected into the discrete sample vials individually, however we have run blanks on combusted amber vials and do not report fluorescence. This will be clarified in the text. Blanks were also run through the melting system into a targeted ultraviolet biological sensor (TUBS) spectrofluorometer, which uses an excitation wavelength of 224nm and collects emission from 280-400nm. All readings of blanks through this unit showed no fluorescence within the 280-400nm emission range, characteristic of dissolved organic material.

3. Line 97. What was the actual absorbance values? These should be plotted as a time series, as A254 is used as a surrogate for DOC in terrestrial systems. It would be interesting for the reader to see this data and for the authors to compare values to other terrestrial systems (e.g. rivers, groundwaters).

All absorbance values were measured below the MQ Water blank run on each day, therefore no values can be used to interrogate the quantity of DOC.

4. Line 106-107. Were the data also processed to remove Rayleigh-Tyndall scatter? How were the Raman and Rayleigh-Tyndall scatter lines processed? Were they replaced by zeros, by NaN (not a number) or was data interpolated? All of these effects can have subtle influence on the resultant PARAFAC model, so it is good to report them.

The EEMS were post-processed to remove the Rayleigh-Tyndall scattering using a MATLAB script of smootheem.m in drEEM version 0.1.0; Murphy et al. 2014. A reference can be added to the text to clarify the smoothing technique to remove each scattering effect.

5. Lines 108-110. The authors must specify what they did for sample classification, normalisation and subset selection. It will be different from Cawley et al (2012), which is just one fluorescence case study, and on pulp mills, so not really very relevant to this research.

We will revise this section to include further details on the procedure for sample classification, normalization, and subset selection prior to PARAFAC modeling. A representative data set was used for PARAFAC modeling, not the entire dataset, so this information will be included to assist others in the same situation.

6. Lines 108-110. Somewhere in this section the authors must quote the value of the standard(s) that they were using. This could be the Raman intensity of Milli-Q water at a specific wavelength, or the intensity of quinine sulphate standards run using the same instrument configuration, or an International Humic Substances Standard, or a tryptophan or tyrosine standard.

The Raman intensity of the MQ Water at a specific wavelength will be provided upon revision.

7. Lines 110-111. More detail is needed on the PARAFAC model, to allow the reader to assess its strength in modelling the data. It is crucial in this paper, as the PARAFAC model is the crux of the whole analysis and interpretation. 1. One would expect to see the core consistency value given. A 'passable' model could be considered have a value of >90%, and a good model a score of >99%. 2. It would be very informative to know why the authors chose a 3 component model over a 2 or 4 component model – did the 4 component model try to model noise, for example? Or did it model a plausible 4th component, but with a low core consistency. 3. The percentage of the data fitted by each component is very valuable information, especially if compared with that from a two and four component model. 4. And finally, a split-half analysis is very useful,

especially if the authors perform a split half analysis using randomly split datasets and a split half analysis with LGM data in one dataset and Holocene data in the other. If the split half analysis fails on the latter test, then it tells you that the LGM and Holocene need different PARAFAC models.

1. The core consistency value can be provided. 2. We will explain the rationale for a 3 component model over a 2 or 4 component model. 3. We can report the percentage of the data fitted by each component to strengthen our argument for a 3 component model. 4. Split half analysis was used for this PARAFAC model and will be reported in the text upon revision.

8. Line 126-127. Amino-acid like fluorescence is too general. Only tryptophan and tyrosine have aromatic groups which fluoresce, and even then, without independent amino acid analyses to confirm their presence, one can never be sure that these compounds are responsible for the fluorescence. If the fluorescence is from an amino acids source, then C1 and C2 look most like a 'tryrosine-like' compound. Tyrosine would excite at both âLij225 nm and âLij275 nm and emit at about 310 nm. But the molecular structure is such that you must observe both the 225 and 275 nm excitation of the 310 nm emission, not just one or the other, as you show in Figure 3. Supplemental Figure 1 confirms the absence of a âLij275 nm excitation peak. Therefore C1 and C2 are not 'tyrosine-like' or 'tryptophan-like'. Model compounds and contaminants that exhibit a single peak in this general region include simple phenols such as cresol (see Aiken, 2014 in Coble et al. (eds) Aquatic Organic Matter Fluorescence), PAHs such as fluorene (Ferretto et al 2014, DOI:10.1016/j.chemosphere.2013.12.087) and aviation fuel (see Baker et al. 2014, Encyc. Anal. Chem. DOI: 10.1002/9780470027318.a9412).

This information will be clarified in the revised text. Chemical composition will be discussed in the manuscript as the fluorescent nature of the OM, along with specific details corresponding to higher/lower molecular weights, aromaticity, reactivity, and potential functional groups identified. Specifically, we will address the potential for monolignol fluorescence as remarked by Aiken in the Aquatic Organic Matter Fluorescence Book

(2014). Appropriate references regarding the fluorescent nature of the potential chemical species present will be included.

9. Line 128. The reference to 'recalcitrant species' is speculative. It would be better to specify the excitation and emission wavelengths of this peak or peaks. I am not aware of fluorescence in this region being recalcitrant – instead bio- and photo- degradation studies show that it is degradable (for example, Osburn et al and Stedmon and Cory, both in Coble et al 2014).

We can correct our usage of 'recalcitrant' in this manuscript, report the excitation and emission wavelengths of the peak/peaks, and cite the appropriate references.

10. Line 129 and Figure 2. There is almost no meaning in 'total OM fluorescence intensities'. Each fluorophore has a different fluorescence efficiency. For example, in this study, you identify three fluorescent components, but each will have a different amount of emitted fluorescence per g C present. So, summing the three is meaningless. It is particularly relevant as low molecular weight compounds such as tryptophan-like and tyrosine-like compounds (argued to be C1 and C2 here) have less chance of their emitted fluorescence being reabsorbed within the molecule, and they therefore have relatively high fluorescence efficiency. In contract, fulvic-like compounds (arguably C3 here) can reabsorb their emitted fluorescence, resulting in a much lower fluorescence efficiency. Figure 2 is therefore just meaningless and instead each PARAFAC component score (C1, C2, C3) needs to be presented.

This is a good point, however, this figure was created to provide a complete record of OM information that tracks relative fluorescent changes of the samples with depth. Removing this figure removes a complete record of all our samples. Using a subset of the samples to build a PARAFAC model created a limitation in the way we can present the depth profile. With our current PARAFAC model, samples were selected as a representative OM character subset of the entire record, not specifically organized to balance how many samples were included in each climate period. It was more informative to

categorize the OM chemical fluorescence into specific groups prior to modeling, rather than to group the climate periods. With statistical outlier testing, it was very challenging to keep a balanced data set in each climate period, thus the most informative results were produced from the categorized subset PARAFAC model. To investigate how the PARAFAC model would shift based on climate periods, three separate PARAFAC models were generated, which produced somewhat redundant results to our original PARAFAC model, and again had large groupings of outliers in some climate periods. However, the changes in PARAFAC component 2 over time were captured using this method, thus added to this work in Figure 3b. These results will be clarified in the text.

11. Line 134 and Figure 3. The PARAFAC scores for C1, C2 and C3 need to be presented in Figure 3. At the moment, no raw data from the PARAFAC model is presented in the paper, yet this is the main focus. The reader has no way of seeing the data and judging its nature e.g. variability over time. Just drawing some PARAFAC model EEMs over an x-y plot would be unacceptable to the fluorescent organic matter research community.

See comment above regarding the subset of samples and how that does not best represent tracking fluorescence intensities over time. We intend to produce a new figure tracking the percentages of each PARAFAC component for our model to show the relative changes. Reporting the 1,191 EEMs would show the variability of fluorescent chemical species over time, however that was unreasonable for this work. The PARAFAC model best represents this variability of the entire record, and then is discussed in terms of the types of chemical nature that is characteristic of each climate period.

12. Line 134-136. This observation needs quantification (see comment above).

That information can be provided upon revision.

13. Line 137-139. As in my earlier comment, you cannot have just one of the two excitation peaks that 'tyrosine-like' compounds excite at, and then call it 'tyrosine-like'.

We can address this by discussing the types of chemical species that would fluoresce in that region. See above comments.

14. Line 139-145. There is most fluorescence at 310 nm, so this is not 'tryptophan-like' at all, as this would also have a peak at 350 nm. More fundamentally, there is a line through the EEM at 310nm which cannot be real. Is this an artefact of the design process of Figure 2, or is it in the actual PARAFAC model? If the latter, it means the model is not correctly modelling the data. Is there anything instrumental e.g. physical filters that change over at 310 nm that could be the cause of this artefact? Is it still present in the 2 component model?

Yes, this feature was present in the 2 component model. The best explanation of this feature is that PARAFAC is doing a great job modelling the EEMs it was given. We acknowledge that challenging qualitative fluorescence data also is modeled as well in PARAFAC, if the samples were not reported as outliers. A discussion on the "unusual" feature of this fluorescence will be discussed in further detail.

15. Line 142. If you performed a single PARAFAC model, then the location of the modelled fluorescence can't change over time. So how can the location of the peak 'move' from LGM to Holocene? Is this from extra PARAFAC analyses that the reader doesn't know about? Or is it a subjective analysis of the original EEMs?

Extra PARAFAC analyses were performed and can be explained more clearly in the text (see comment above for details). It was not a subjective analysis of the original EEMs.

16. Line 145-150. I would disagree with this interpretation. This fluorescence is typical of 'peak A' and 'peak C' compounds. A peak 'M' fluorescence would be blue shifted compared to 'peak A', and in your component C3 there are two peaks and they both have the same emission wavelengths.

The manuscript will be revised to correct for peaks A and C result reporting and discussion of Holocene OM chemical fluorescence.

17. Lines 153-155. The fact that no one else has reported your fluorescence peaks is either very exciting or very worrying. It would suggest that what you are seeing is not anything that has been reported before e.g. you are not seeing 'tyrosine-like' fluorescence, and by implication, you can't definitively interpret it as a microbial signal.

Of the data available in the OpenFluor database, a repository of a selection of samples (not every fluorescent study completed), our results showed no matches with other PARAFAC components. This is reasonable given the scope of the project and the great volume of samples spanning 6,000 to 27,000 years ago from ice. Yes, we agree that what we are seeing is not anything that has been reported before. We also agree that your suggestion as the correct interpretation of the PARAFAC components would not distinctly be tyrosine-like or tryptophan-like, thus the interpretation of a microbial signal is not definitive. These interpretations will be edited accordingly upon revision.

18. Line 160 and Figure 4. The authors state that Figure 4 shows the 'PARAFAC components', but there is just one line. What is this? Is it C1, or C2, or C3? All three components must be shown individually, here and in Figure 3.

The PARAFAC components determined in each climate period are provided on the same graph for your convenience. The black line refers to the $\delta$18O record. This figure will be edited considerable to show the nssCa and $\delta$18O record only.

19. Line 175. C1 and C2 PARAFAC model scores need to be plotted in Figure 3. Line 184. C3 PARAFAC model scores need to be plotted in Figure 3.

Model scores can be provided.

20. Line 189-191. This observation is unremarkable, as all humic and fulvic substances standards have a higher fluorescence intensity at the short excitation wavelength (see examples in Aiken (2014)).

Correct. We can adjust this appropriately and provide the reference.

21. Line 191. It sounds like you are saying that there are plants and soil in the ice? I'm sure you don't mean that?

Yes, thank you. This was an error in phrasing and can be corrected upon revision.

22. Line 214. No fluorescence data over time is presented (except for the total fluorescence, which is not meaningful). So this section is speculative.

Correct. Please see comments above addressing our explanations and routes for revision.

New fluorescent figures (Figs 2 and 3) are provided for consideration.

———————————————————

[Figure]

Figure 2. PARAFAC analysis results for West Antarctic Ice Sheet Divide ice core organic matter showing a) components 1, 2, and 3 (C1, C2, and C3) and b) the fluorescence percentage of each component contributing to the overall fluorescence signature over the Last Glacial Maximum (LGM), last deglaciation (LD), and Holocene climate periods as a function of time (kyr before present 1950). Average fluorescence percentages (gray dashed lines) are provided for each component, separately calculated for each climate period. Fluorescent data were reported in Raman Units. Note: C3 average fluorescence percentages ranged from 0-2% in the LGM and LD, and did not correspond to resolved fluorophores.

**Fig. 1.** Figure 2. PARAFAC analysis results for West Antarctic Ice Sheet Divide ice core organic matter showing a) components 1, 2, and 3 (C1, C2, and C3) and b) the fluorescence percentage of each component c

[Figure]

Figure 3. PARAFAC analysis results of component 2 (C2) variation with climate periods a) Last Glacial Maximum (LGM), b) last deglaciation (LD), and c) Holocene. Components 1 and 3 from the PARAFAC model in Figure 2 showed no variability over time.

**Fig. 2.** Figure 3. PARAFAC analysis results of component 2 (C2) variation with climate periods a) Last Glacial Maximum (LGM), b) last deglaciation (LD), and c) Holocene. Components 1 and 3 from the PARAFAC mo

---

## Author Response (AR1)

**RC1**

The reviewer comments are numbered for reference. Each reply is listed below the numbered reviewer comment.

1. Title and text. The term "organic matter quality" seems to be not adequate to describe the measurements here reported. Really, just fluorescence measurements were carried out and interpreted as signatures of some classes of organic components-like markers. I'd suggest the term "organic matter markers" or "organic fluorescent components".

Fluorescence measurements were carried out and interpreted as signatures of organic components. The chemical nature of the fluorescent fraction of the organic matter was surveyed using a fluorescent technique, thus organic matter markers is an appropriate alternative for the title and text. The title has been edited to, "A 21,000 year record of fluorescent organic matter markers in the WAIS Divide ice core". The text has been edited accordingly throughout the manuscript to improve clarity.

2. Line 17 and several other points. Usually, time unit is expressed as "kyr" and not as "kyrs". Please, correct in the text and figures.

All "kyrs" have been corrected to "kyr."

3. Lines 20-22. Here or in the "Results" section, Authors should clarify what They mean with the terms "labile microbial OM", "recalcitrant OM", "bioavailable carbon species" etc. A very short description of these terms could help the reader in better understanding the different biological significance and the different availability in carbon exchange between cryosphere and other ecosystems.

The descriptions of labile (easily altered) and recalcitrant (less easily altered) OM marker descriptions have been added to the abstract and the text.

4. Line 32. Please, cite also Wolff et al., Southern Ocean sea-ice extent, productivity and iron flux over the past eight glacial cycles. Nature, 2006, Vol. 440, 491-496, doi:10.1038/nature04614.

This reference was included upon revision in Lines 36-39.

"Numerous inorganic species trapped in ice has been used to reconstruct past chemical compositions of the atmosphere, its recent change in response to growing human activities as well its past natural variability (Legrand and Mayewski, 1997; Petit et al., 1999; Johnsen et al., 2001; Alley, 2002; Wolff et al., 2006; Jansen et al., 2007; Luthi et al., 2008)."

5. Lines 51 and following. What "OM character" means? Chemical composition? Chemicalspecies or functional groups identification? Authors are requested to clarify their thought.

The text has been edited for clarity, including the usage of OM markers in Lines 54-61. "While still a novel addition to deep ice core research, chemically characterizing the OM markers (i.e.

composition and chemical species) in englacial ice is of particular interest for several reasons: 1) OM markers can be linked to its source (e.g., aquatic, terrestrial) describing different influences of past and present ecosystems, 2) OM markers can serve as a proxy for englacial biological activity from *in situ* production, potentially explaining anomalous concentrations of other gases (e.g., methane, carbon dioxide) in ice core research, and 3) OM could be a pivotal contributor to the global carbon cycle if materials released to surrounding environments are metabolized to greenhouse gases (e.g., carbon dioxide, methane) in a warming climate."

6. Line 55. Even methane formed in anaerobic conditions is a strong forcing factor in the warming climate.

We describe the release of organic material upon melting of polar ice and the potential for it to be metabolized to greenhouse gases, such as carbon dioxide and methane in Lines 54-61. "While still a novel addition to deep ice core research, chemically characterizing the OM markers (i.e. chemical species and composition) in englacial ice is of particular interest for several reasons: 1) OM markers can be linked to its source (e.g., aquatic, terrestrial) describing different influences of past and present ecosystems, 2) OM markers can serve as a proxy for englacial biological activity from *in situ* production, potentially explaining anomalous concentrations of other gases (e.g., methane, carbon dioxide) in ice core research, and 3) OM could be a pivotal contributor to the global carbon cycle if materials released to surrounding environments are metabolized to greenhouse gasses (e.g., carbon dioxide and methane) in a warming climate."

7. Line 72. Since snow density is variable, it is better to express the mean accumulation rate as cm or mm "water equivalent".

This has been edited in the text to an average annual accumulation rate  $0.207m_{weq} a^{-1}$  (Banta et al., 2008) in Lines 78-81.

"Snow precipitation at this site is relatively high with an average annual accumulation rate  $0.207 m_{weq} a^{-1}$  (Banta et al., 2008)..."

8. Lines 74-76. What means this sentence? Several other ice cores (for instance, Taylor Dome and Talos Dome, in the same Antarctic Sector; Dome C and Dome Fuji, in the inner Antarctica; Dronning Maud Land, in the Atlantic Sector; etc.), even drilled before WD ice core, constitute "equivalent paleoclimate record" to Greenland ice cores. In particular, the EDC, EDML and DF climate records were compared with the climate oscillations recorded along the NGRIP ice core in: EPICA Community Members, Oneto-one coupling of glacial climate variability in Greenland and Antarctica. Nature, 2006, Vol. 444, 195-198, doi:10.1038/nature05301.

The statement was deleted upon revision.

9. Line 80. Please, change "drilling solvent" with "drilling fluid".

"Drilling solvent" was edited to "drilling fluid" in the text.

10. Line 88. Please change "combusted" with "pre-fired".

"Combusted" was be edited to "pre-fired" in the text.

11. Section 2.3. The correction for the absorbance measurements seems to be not clear. Authors are asked to give more information on that. Besides, the absorbance threshold seems to be quite high. If a.u. means, as I think, absorbance unit, the value A = 0.3 corresponds to a percentage transmittance of 50% (A = Log 1/T) that seems to be too low for ice-core melted water at 254 nm. Maybe, some particles were suspended or some gas bubbles were present in the melted samples during the measurements. Authors are requested to clarify this point.

The text was edited to reflect that all of our samples were optically transparent in Lines 100-109. "Prior to fluorescence spectroscopy, absorbance spectra of WD core meltwater samples were collected from 190-1100 nm (UV-Vis spectral range) using a Genesys 10 Series (Thermo-Scientific) Spectrophotometer with a 1 cm path length cuvette and VISIONlite software. Obtaining UV-Vis absorbance spectra are necessary for the post-processing calculations of spectral corrections including primary and secondary inner filter effects (Acree et al., 1991; Tucker et al., 1992). Absorbance values at 254 nm (A254) greater than 0.3 absorbance units (a.u.) require dilution prior collecting the UV-Vis absorbance spectra and EEMs. WD core OM samples were optically transparent, with measured A254 values well below 0.3 a.u. after blank correction; consequently, no sample dilution prior to UV-Vis absorbance measurements and EEMs was required (Miller and McKnight, 2010; Miller et al., 2010). Spectra were blank corrected against purified water from a Milli-Q system each day. UV-Vis absorbance spectra were subsequently incorporated into the spectral corrections calculations for post-processing the EEMs data."

12. Line 97. Maybe the term "optically dilute" could be changed in "optically transparent" (but I do not think that this term is correct for T% = 50%).

The term "optically transparent" was edited in the text.

13. Section 2.4. Even if a reference is cited, Authors are requested to give some basic information about the PARAFAC multivariate analysis.

Basic information on the setup of our analyses and specific details regarding PARAFAC was added to the manuscript in Lines 123-151.

[revised manuscript text omitted]

15. Lines 126 and 128. Authors are requested to shortly describe the characteristics of "bioavailable carbon species" and "more recalcitrant species".

The text has been edited to include such definitions in Lines 167-173.

"All samples contained low Ex/Em wavelength (240-270 nm / 300-350 nm) fluorescence characteristic of more easily altered material by microorganisms, representing fluorescent OM markers potentially of proteinaceous (Coble et al., 1990; Coble et al., 1998), polycyclic aromatic hydrocarbon (PAH) (Ferretto et al., 2014), and simple phenol, tannin, or monolignol (Coble, 2014) origin. Fewer samples (2.5%) contained OM fluorescence at higher Ex/Em wavelengths (240-250 nm / 340-530 nm), characteristic of more humic-like markers of terrestrial plant/soil origin (Coble et al., 1990). Examples of low and high Ex/Em wavelength fluorescence can be seen in Supplement Figure 1a-c."

16. Lines 129-131. This early Holocene peak of fluorescent mater is interesting, as well as the larger peak around 21-22 kyr BP. Authors do not discuss these two features in the temporal profile of the WD ice core. I'd like to know the Author interpretation on these large depositions of organic fluorescent compounds, even if as a tentative hypothesis. It should be very interesting to perform some qualitative analysis (e.g., by HPLC-MS measurements) on these samples in order to clarify the nature of the fluorescent compounds.

The fluorescent peaks were discussed prior to the revised manuscript as intensity shifts in each climate period, and do not directly correspond with large depositions of organic fluorescent compounds. Rather, the quantum yields of specific fluorescing material is represented, along with the hypotheses that both fluorescing material and concentration of organic material may be contributing to shifts in fluorescence intensities. Large deposition events of organic material cannot be linked to shifts in fluorescence intensities. Further analyses of these samples is not possible as only 7.5mL of each sample were available and have been used in the present analyses. This figure no longer accurately represents our data set and was removed upon revision.

17. Lines 132 and following. I surely do not want to minimize the contribution of the PARAFAC analysis, but I have to note that the result of its application is quite basic. From Figure S1, the separation of the fluorescent bands at 420 nm Em and 300 nm Em is very clear even without any multi-parametric analysis. The only significant result is the identification of two fluorescent components C1 and C2 at short Em and Ex wavelength. However, the two components are just attributed to two large organic compound classes (amino acid-like fluorescent compounds), without a more specific characterization. Besides, the C1 and C2 fluorescent components are not clearly differentiated in terms of biological origin: C1 is attributed to tyrosine-like fluorescent compounds associated to "microbial processing in aquatic environment", while C2 is described as a fluorescent signature overlapping "between tyrosine- and tryptophan-like" fluorescent compounds. At line 177-178, Authors just report that C2 containing tryptophan-like fluorescence could represent "intact dissolved proteins ….freshly derived from microorganisms". Authors are requested to better organize, in the present section, the discussion on the possible origin of these

components and to enlighten the biological and environmental differences. In conclusion, the PARAFAC analysis seems to be not able to "resolve the representative subset of samples into individual OM fluorescing components", as the Authors assessed at lines 132-133. Even the comparison with the OpenFluor database components did not give significant matches (if I have well understood lines 153-155).

RC1 comment: The separation of the fluorescent bands at 420 nm Em and 300 nm Em is very clear even without any multi-parametric analysis.

Author's reply: This is correct, however Figure S1 highlights examples of different types of fluorescing organic matter, so it was our intention to show notably obvious differences from the WD core. The WD core fluorescent data set comprised a small fraction of material fluorescing in Figure S1 (c), thus we needed to apply a statistical tool, PARAFAC analysis, to decompose the EEMs into individual fluorescing components, even for the overlapping fluorescing material at lower Ex/Em wavelengths. The significant result is that PARAFAC was used as a multiparametric tool to decompose the EEMs data set into three fluorescing components. That information was subsequently categorized to identify the chemical character of the fluorescing organic material characteristic of each climate period. RC1 Comment: "Besides, the C1 and C2 fluorescent components are not clearly differentiated in terms of biological origin: C1 is attributed to tyrosine-like fluorescent compounds associated to "microbial processing in aquatic environment", while C2 is described as a fluorescent signature overlapping "between tyrosine-and tryptophan-like" fluorescent compounds." C1 and C2 fluorescence technique. The specific characterization of C1 has been revised in Lines 180-196 as:

"Three OM PARAFAC components were identified from the WD EEMs (fluorescing regions shown in Figure 2a, and Ex/Em wavelength loading scores shown in Supplemental Figure 2). PARAFAC component one (C1; Figure 2a, top) showed maximum fluorescence in a region analogous to the secondary fluorescence of fluorophore peak B (tyrosine-like, Ex: 240 nm and Em: 300 nm), typically associated with microbial processing in aquatic environments (Coble et al., 1990; Coble et al., 1998). Regions of fluorescence at such Ex/Em wavelengths are commonly referred to as "protein-like" but overlap with fluorescence of other origins (Coble, 2014). However, without the primary region of fluorescence associated with fluorophore peak B (tyrosine-like) at higher Ex/Em wavelengths, the OM fluorescent marker of C1 cannot be determined to be tyrosine-like material of microbial origin by this method. Rather, OM with similar Ex/Em wavelength fluorescence has been documented for simple phenols (e.g., tannins and monolignols) commonly detected in natural waters (Coble, 2014). Simple phenolic OM is characteristically lower in molecular weight, aromaticity, and is considered to be more easily altered in the environment, as compared to more humic-like material (Coble, 2014). Thus, we report the chemical composition of WD OM in C1 to be most similar to monolignol chemical species, ubiquitously found in the environment as the precursors to lignin material detected in vascular plants. Once thought to be generated in the environment from tyrosine, the biosynthesis of monolignols actually originates from phenylalanine via multiple enzymatic reactions, therefore sharing protein-like origin, but ultimately is chemically linked to vascular plants as a fluorescent OM marker (Wang et al., 2013)."

The specific characterization of PARAFAC component two (C2) has been revised in Lines 197-208 to:

"PARAFAC component two (C2; Figure 2a, middle) contained maximum fluorescence at low Ex/Em wavelengths (260-270 nm / 310-320 nm) in regions analogous to the primary fluorescence of fluorophore peak B, and cresol (methylphenol), commonly known as the building blocks of tannins (Kraus et al., 2003), the major components of soil and aquatic humic OM (Tipping, 1986). Secondary fluorescence commonly detected for fluorophore peak B (tyrosine-like) was not observed for C2, and the combination of fluorescence from C1 and C2 do not yield the appropriate primary and secondary fluorescent trends commonly associated with tyrosine-like OM. Therefore, by this method, PARAFAC identified two distinct components, that may have protein-like similarities, but cannot be inherently linked to amino acid-like material and microbial origin. Thus, we determined that C2 fluorescence was characteristic of a combination of protein-like and tannin-like OM markers based on the regions of overlapping fluorescence by this method. Similarly to the chemical species reported for C1, the low Ex/Em wavelength fluorescence of C2 indicates OM markers with lower molecular weights, aromaticity, and chemical species that are more easily degraded in the environment by microorganisms (Coble, 2014)."

The specific characterization of PARAFAC component three (C3) has been revised in Lines 213-226 to:

"Component three (C3) displayed fluorescence commonly associated with more humic-like material. Two humic-like fluorescing regions were identified that comprised C3: fluorescence at 1) Ex/Em: 240-260/380-460 nm, characteristic of fluorophore peak A, and 2) Ex/Em: 300-320/380-460 nm, characteristic of fluorophore peak C, commonly associated with terrestrial plant and/or soil origin (Coble, 1996; Marhaba et al., 2000). Fluorescent OM markers in this region is linked with chemical species having higher molecular weights aromatic nature, and are considered to be less easily altered by biodegradation in the environment as compared to more labile material (Coble et al., 1990; Cory and McKnight, 2005; Murphy et al., 2008; Balcarczyk et al., 2009; Fellman et al., 2010). While commonly referred to as the "more recalcitrant" fraction of fluorescent OM, studies have shown that terrestrial humic-like material is susceptible to photodegradation, therefore should not be considered as an unalterable fraction of OM (Osburn et al., 2001; Stedmon et al., 2007)."

More specific characterizations of each PARAFAC component cannot be determined using this bulk analytical technique, however possible chemical species were suggested. This was clarified in the text. The chemical species associated with PARAFAC C1, C2, and C3 were discussed in the text providing a bulk representation of the organic materials present throughout 21,000 years. This sets the foundation for future work, a point which was also be clarified in the text in Lines 286-288.

"The composition and chemical origins associated with PARAFAC components C1, C2, and C3 provided a bulk level representation of the OM markers present throughout 21.0 kyr and initiated the foundation for future research."

RC1 Comment: "Authors are requested to better organize, in the present section, the discussion on the possible origin of these components and to enlighten the biological and environmental differences." The discussion on the possible origin based on these data is present in the manuscript, highlighting environmental differences over time. RC1 Comment: "In conclusion, the PARAFAC analysis seems to be not able to "resolve the representative subset of samples into individual OM fluorescing components", as the Authors assessed at lines 132-133. Even the comparison with the OpenFluor database components did not give significant matches (if I have well understood lines 153-155)."

The authors disagree. The PARAFAC analysis resolved the representative subset of samples into the only individual OM fluorescing components that were present in the samples. PARAFAC analysis is capable of producing brilliant results of the data set asked of it. With most EEMs resulting in the example provided in Figure S1, it was not surprising to have the low Ex/Em wavelength fluorescent components modeled as two individual components C1 and C2, prior to C3. The order of the modeled components describes the variation in the data set, and was statistically validated with the drEEM program in MATLAB. The OpenFluor database contains various data sets from samples collected around the world. Submission of data to OpenFluor is not a requirement, and is currently still in its growing phases. Scientists are encouraged to upload their PARAFAC datasets upon publication, but it is not required, thus the database it contains does not encompass all possible fluorescent component data. OpenFluor matches with the dataset describe PARAFAC components that have been identified in other ecosystems. A match or no match result describes unique data worth reporting. We felt it was interesting to report that organic material from 6,000 to 27,000 years ago did not match any of the uploaded PARAFAC data currently in the database. Our dataset is the first of its kind from a continuous Antarctic ice core, thus we stress the importance of its upload to OpenFluor upon publication, which in turn will better serve the fluorescent community.

The text has been edited to incorporate our rationale behind the results of the OpenFluor query in Lines 227-233:

"WD ice core OM PARAFAC components were uploaded to the OpenFluor database to compare and contrast C1, C2, and C3 with other environmental OM marker studies, however, no component matches were determined (Murphy et al., 2014). The OpenFluor database is a repository of a selection of samples, and while still growing to encompass a thorough library of fluorescent OM markers from highly variable environments, it is reasonable to expect nonmatching results based on database queries. Our results matched no previously identified PARAFAC components uploaded to the database, which we attribute to the unique scope of this work and the great volume of samples spanning 21.0 kyr from Antarctic ice."

18. Lines 142-143. The terms "red/blue shifted to longer/shorter Em wavelengths" are repetitions. Please, change in "Em-wavelength red/blue shifted" or "shifted to longer/shorter Em wavelengths". Authors should clarify the statistical significance of these shifts (especially from LGM to LD) and anticipate the consequent biological meaning (especially from LGM-LD to Holocene). Besides, which is the meaning of the red or blue shifts? When blue (red) shift occurs, is the C2 component a marker of tyrosine-like (tryptophan-like) fluorescent compounds?

The text has been edited accordingly and descriptions of chemical species associated with the specific shifts are provided in Lines 260-265.

19. Section 3.2. The relationship between glacial cycles and atmospheric deposition of dust in Antarctica is a very relevant and largely discussed topic in ice core studies. Here, the Authors have to take for granted the inverse relationship between site temperature and dust deposition (by citing the most relevant references) and anticipate the discussion on the possible relationships among temperature, dust and biological activity (or OM transport efficiency), as revealed by the

fluorescence temporal profile. At this purpose, Authors should choose the preferred dust indicator among the possible dust markers measured along the WD ice core (nss-Ca, Mn and Sr), also basing on the correlations between the elements (lines 165-166).

The authors agree and highlight our speculative relationships to better infer the influences of continental dust loading on OM markers. nssCa was selected to be our preferred dust indicator, highlighting that the concentrations agree with other ice core records, and then discuss the proxy in terms of OM influence over time (Lines 316-323).

We incorporated these statements as:

"Concentrations of nssCa has been shown to be a valuable proxy for terrestrial crustal dust in paleoclimate ice core records (McConnell et al., 2007; Gornitz, 2009; Lambert et al., 2012). As such, it is plausible to envisage a link between the concentrations of nssCa and other transported materials influenced by aeolian deposition, (e.g., OM concentration and character, microbial biomass, and pollen grains). The relationship between glacial cycles and atmospheric deposition of dust in Antarctica is largely discussed in ice core studies. We applied an assumption that common transport processes of dust and OM markers together could be hypothesized only if dust and OM originated from the same continental areas. Therefore, in this work, we merely speculate on the influence of dust concentrations and OM composition measurable by fluorescence spectroscopy."

20. Lines 174 and 176. Maybe, "throughout time" is better than "throughout history".

The text has been edited to reflect this addition throughout the manuscript.

21. Line 198-200. Common transport processes of dust and OM could be hypothesized only if dust and OM originated from the same continental areas. In LGM, Southern South America was supposed to be the major dust source area for Antarctica. In LD and, especially, Holocene, even Australia could have played a significant role. Therefore, Authors implicitly suppose that OM was originated in these continental regions. For OM originated by marine sectors (C1, C2?, part of C3), the relationship with dust transport processes cannot be considered significant because they can follow very different pathways (e.g., implying different meridional or zonal atmospheric circulation modes).

Common transport processes of OM were only hypothesized, and we inferred a local South American major dust source region for Antarctica. The text was edited throughout the discussion sections to reflect these revisions.

22. Lines 200-201. Authors here refer on relationships between dissolved organic carbon and dust markers. I suppose DOC measurements were not performed as part of this paper (see following sentence in the text). Authors should give more information on that or cite some reference.

DOC concentrations were not performed as part of this work. The supplemental information section outlining these relationships were removed upon revision.

23. Line 204. I think Authors refer to Figure 4.

Indeed, we did. Thank you. The manuscript has been edited.

24. Lines 205-212. This part has to be completely revised. The complex relationship between dust deposition in Antarctic ice cores and climatic cycles cannot be discussed in this form in this paper and, how I have already pointed out, has been (and will be) the topic for several specific papers. Authors are requested to report the major literature references about LGM-LD-Holocene dust/climate pattern and focus the discussion on the relationship among climate, dust (possibly) and OM fluorescent markers. Besides, I have to note that the detail in the discussion on the behavior of OM data and dust profiles along the WD ice core is not so high to appreciate specific differences in nss-Ca, Mn and Sr profiles. Therefore, since the three dust-marker profiles were not singularly discussed and differentiated, I'd suggest to replot Figure 4 with just one dust marker (maybe, nss-Ca).

One dust marker (nssCa) is now presented and only discussed in terms of the fluctuations of dust concentrations with the different OM chemical species present in different climate periods. This section was completely revised to reflect the changes and a new Figure 4 is now incorporated (see below with caption).

Figure 4. Trace element concentration of (top) non-sea salt calcium (nssCa; ppb), and (bottom) the  $\delta$ 18O (per mil) temperature record (Marcott et al., 2014) from the West Antarctic Ice Sheet Divide ice core as a function of time (kyr before present 1950), dating from the Last Glacial Maximum (LGM), through the last deglaciation (LD), to the mid-Holocene.

25. Section 4.3. Even this section has to be largely revised. Authors assume a series of speculations to correlate changes of OM fluorescent markers to changes in climatic and environmental conditions, as evaluated by changes in sea-ice coverage (by ss-Na – Authors could add the ss-Na profile in figure 4), dust production and transport (by dust markers) and volcanic eruption frequency (by nss-SO4 spikes) in the LGM, LD and Holocene. However, no reliable comparison among the different time profiles is shown. In particular, while dust and sea ice markers show a progressive decreasing during the LD, the OM fluorescent profile shows an abrupt change (at about 18.5 kyr BP) from high LGM values and very low LD and Holocene levels. All the discussion is too elemental and also the changes in C1 and C2 relative contributions are not clearly interpreted. From the data here reported, I can just see that OM fluorescent markers are high in the LGM, when dust and sea spray are high. However, there is not experimental evidence on which climatic or environmental factors (more efficient meridional or zonal atmospheric transport, larger sea ice coverage, higher input from continental areas, larger emissions from marine biota, etc.) could have driven the OM deposition at the WD site. Finally, the relationship between volcanic activity (as recorded by the nss-SO4 spikes along the WD ice core) and OM fluorescent markers is, in my opinion, really unsustainable. Volcanic signatures in Antarctic ice core are mainly related to long range atmospheric (especially stratospheric) transport of SO2 emitted during eruptions occurred at hemispheric scale and it is really difficult to correlate changes in WD OM to sporadic, short-time and widespread volcanic emissions without a strong experimental evidence.

The authors agree. The co-registered geochemical WD dataset were used to speculate on the origin of the OM characterized by fluorescence spectroscopy. No direct comparisons were reported because none were available for this project; that was beyond the scope of this work. The manuscript has been edited to reflect this information in Lines275-278. "It is important to note no direct comparisons between dust concentrations and OM qualitative markers or concentrations can be made with these data, as that was beyond the scope of this work. Rather, this information was subsequently utilized as discussion points to infer more information regarding the OM marker origin detected in the WD ice core."

The PARAFAC components C1 and C2 relative contributions were discussed in terms of percentages relative to the other components. See above responses outlining our PARAFAC component discussion section in further detail. The sections outlining the volcanic signatures is highly speculative and was removed upon revision.

26. Lines 225-226. What this sentence means? What is compared to the open ocean?

The authors meant to state a comparison of more to less sea-ice extent. This was edited in Lines 350-351.

27. Lines 233-234. Authors are requested to better discuss the red shift of the C2 component, explaining which amino acid-like components increases its contribution to fluorescent OM and at which biological source can be attributed. What "external environments" means?

The red shift clarification was added with chemical species clarifications as stated above. "External environments" was deleted upon revision of the manuscript.

28. Lines 237-243. The pattern of the OM fluorescent markers during the ACR is not visible in Figure 3 (neither in Figure 2). This part is merely speculative and not supported by experimental evidences.

Correct. The dust record was used in Figure 4 as a discussion point to speculate on the variation in OM character during the ACR, specifically for the variability in PARAFAC C2 transitioning from LGM, through the LD, to the Holocene. The text was edited to reflect these revisions in Lines 367-372.

"The years between 13.0-11.5 kyr BP, at the end of the LD, defined as the Antarctic cold reversal, incorporate a depression of temperature, just prior to the early Holocene, where reports of Ca and dust concentrations increase (Delmonte et al., 2002). Also measured in the WD ice core (increases in nssCa; Figure 4), we speculate that these environmental fluctuations during the Antarctic cold reversal, may also explain the fluorescent OM variation in the LD (Figure 3a-c)."

29. Lines 244-250. How can the Authors explain the very low levels of OM fluorescent markers during the Holocene, when climatic conditions should promote higher terrestrial and marine biological productivity? Which could be the significance of the large spike in OM fluorescent profile (Figure 2) at about 10 kyr BP?

This section was completely revised and Figure 2 was removed upon revision. We cannot discount that carbon productivity is reportedly higher in the Holocene, however, that does not ensure efficient transport of materials to Antarctica. We thus reported our findings and discuss these ideas with the need for future investigations that could answer such questions.

30. Line 258. Please, change "Concentrations of nss-sulfur..." with "Spikes in nss-SO4 concentrations..."

The sections outlining volcanic activity discussed in terms of OM character were removed.

31. Lines 261-262. Authors are requested to clarify how volcanic activity can stimulate OM production. How is calculated the percentage of the fluorescent OM attributed to the volcanic activity? The relationship between volcanic activity and OM deposition at WD site is, in my opinion, not plausible and not supported by experimental data (at least, by experimental data here reported). Have the Authors measured OM fluorescent peaks in ice core sections with volcanic depositions? In absence of experimental support, the discussion about the volcanic activity and OM fluorescent markers should be removed from the manuscript.

Indeed, we do not have experimental support, merely just speculations on this topic. Volcanic activity discussion sections were removed upon revision.

32. Conclusions section. This part should be changed accordingly to the changes suggested along the different manuscript sections.

This section was revised accordingly based on all the reviewer's comments. RC2

The reviewer comments are numbered for reference. Each reply is listed below the numbered reviewer comment.

1. You state that "Ice core studies rely on the paradigm that atmospheric deposition is the sole mechanism for specific gases and materials to become trapped in the ice" (Lines 47- 48) yet it is unclear if you apply this paradigm to your work. If you do not allow even a remote chance for insitu production of this organic matter, then please explicitly state so in your work.

We apply this conservative approach for organic matter preservation in ice cores, however, do so acknowledge the possibility for *in situ* production of organic matter. This acknowledgement is addressed in Lines 54-61.

"While still a novel addition to deep ice core research, chemically characterizing the OM markers (i.e. composition and chemical species) in englacial ice is of particular interest for several reasons: 1) OM markers can be linked to its source (e.g., aquatic, terrestrial) describing different influences of past and present ecosystems, 2) OM markers can serve as a proxy for englacial biological activity from *in situ* production, potentially explaining anomalous concentrations of other gases (e.g., methane, carbon dioxide) in ice core research, and 3) OM could be a pivotal contributor to the global carbon cycle if materials released to surrounding environments are metabolized to greenhouse gases (e.g., carbon dioxide, methane) in a warming climate."

We can only speculate on the possibility of *in situ* organic matter production due to methodological limitations and thus, that clarification is edited in the text in Lines 67-70. "For this study, all possible sources of OM markers detected within the West Antarctic Ice Sheet (WAIS) Divide ice cores were considered, however we can only speculate on the possibility of *in situ* OM production due to methodological limitations."

2. In lines 179-183 you mention the possibility of in situ OM processing but then do not discuss if such transformation could affect the samples in this work.

The detection of tyrosine-like and tryptophan-like organic material would mark the signs of microbially derived OM produced in the WD ice core. However, it would also mark the signs of microbially derived transported material to the WD ice core. Due to methodological limitations and the lack of protein-like resolved fluorophores, we can only mentioned the possibility of in situ processing in the WD ice core. Future work in deep ice will have to be conducted to specifically test for such occurring transformations/identification of microbially produced OM. This section was edited accordingly in Lines 294-305:

"We continue to acknowledge the fluorescence overlap at low Ex/Em wavelengths (C1 and C2) with regions of fluorescence that describe "protein-like" material characteristic of microbial processes (*in situ* OM production and transformation), however cannot confirm OM microbial origin by fluorescence spectroscopy alone. Miteva et al. (2016) reported that the presence of microorganisms in deep ice cores also suggests the possibility of *in situ* OM processing, which could have important implications for gaseous climate records (Rhodes et al., 2013; Miteva et al., 2016; Rhodes et al., 2016). Including *in situ* biological OM transformations in ice core research was recently proposed as an alternate mechanism for CH4 production in ice from firn layers of the WD ice core (Rhodes et al., 2016). At this juncture, our bulk level fluorescent results cannot

argue that *in situ* OM production is a major contribution to the OM markers detected by EEMs in the WD ice core due to the lack of resolved tyrosine- and tryptophan-like fluorescing components."

3. You mention that tryptophan-like florescence in C2 may derive from microorganisms, and then mention that the presence of microorganisms may result in in situ OM processing, but step back from linking the two aspects.

This section was completely revised and no longer should include a discussion of the linkage between C2 and in situ production.

4. In the following paragraph you then mention that Holocene terrestrial plants and soils are the likely source of the C3 OM yet do not mention if in situ processes may affect this material or if you ascribe this material to be solely brought in via atmospheric transport. Please clarify your stance on the source and possible post-depositional processes affecting the samples as both aspects are essential to your interpretations of the data.

Upon clarification of the points mentioned above from the previous comment, this section was edited accordingly.

5. Please check that all figures are cited in the text. In lines 128-144 you mention Supplemental Figures 1a-b. You do not refer to Figure 2 in the text. As you refer to the Supplemental Figures but not Figure 2, then perhaps their roles should be reversed with the current Figure 2 included in the Supplementary Information and vice versa.

Figure 2 was cited in the old manuscript Line 129. All figures are cited appropriately in the revised manuscript.

6. The left bars and corresponding explanation in the caption of Figure 3 are confusing. In the article text you explicitly state that C3 only occurs during the Holocene. As most readers will likely first look at the figures and captions before reading the article, it bears mentioning in the caption that C3 is specific to the Holocene. Demonstrating the variation in C2 by various time periods (LGM, LD and Holocene) is useful but then makes the reader immediately wonder what is the variation in C1 between climate periods. If there is no substantial variation between time periods for C1, please mention this fact in the caption.

New figures 2 and 3 are incorporated into the revised manuscript. Supplement Figures describing the separate PARAFAC model are also included to highlight that variation in C2 was the only component shifting with climate. These figures were discussed appropriately in the text. Figure 2 and caption: